# Cell-fate conversion of intestinal cells in adult *Drosophila* midgut by depleting a single transcription factor

Xingting Guo[1,2], Chenhui Wang [1,3] ✉, Yongchao Zhang[1,2], Ruxue Wei[1] & Rongwen Xi [1,2] ✉

The manipulation of cell identity by reprograming holds immense potential in regenerative medicine, but is often limited by the inefficient acquisition of fully functional cells. This problem can potentially be resolved by better understanding the reprogramming process using in vivo genetic models, which are currently scarce. Here we report that both enterocytes (ECs) and enteroendocrine cells (EEs) in adult *Drosophila* midgut show a surprising degree of cell plasticity. Depleting the transcription factor Tramtrack in the differentiated ECs can initiate Prospero-mediated cell transdifferentiation, leading to EE-like cells. On the other hand, depletion of Prospero in the differentiated EEs can lead to the loss of EE-specific transcription programs and the gain of intestinal progenitor cell identity, allowing cell cycle re-entry or differentiation into ECs. We find that intestinal progenitor cells, ECs, and EEs have a similar chromatin accessibility profile, supporting the concept that cell plasticity is enabled by pre-existing chromatin accessibility with switchable transcription programs. Further genetic analysis with this system reveals that the NuRD chromatin remodeling complex, cell lineage confliction, and age act as barriers to EC-to-EE transdifferentiation. The establishment of this genetically tractable in vivo model should facilitate mechanistic investigation of cell plasticity at the molecular and genetic level.

It was a longstanding view that terminally differentiated somatic cells were stable and could not be changed. However, in recent decades, it has been found that the forced expression of a cocktail of transcription factors (TFs) or small molecules can alter the developmental identity of a cell into another type[1,2]. Moreover, the reprogramming process occurs naturally in somatic tissue injury, such as the dedifferentiation of committed progenitor cells or differentiated cells back to stem cells in the injured airway and intestine[3,4]. Somatic cell fate plasticity provides great potential in regenerative medicine, and the concept of in situ cell fate reprogramming, which aims to convert resident cells into the desired cells within the tissue, is currently under exploration

for the generation of various cell types, such as neurons, cardiomyocytes, and β-cells[5–8]. However, these approaches are often limited by inefficient cell reprogramming that cannot acquire sufficient function of the desired cell types, which is at least in part due to our incomplete understanding of the molecular mechanisms underlying cell reprogramming and cell identity maintenance.

The intestinal stem cell (ISC) lineage in the *Drosophila* midgut is an ideal system for studying cell fate specification and maintenance. It is due to its similar yet less complex cellular composition and regulatory mechanisms, relative to the mammalian intestine, as well as the advantage in genetic manipulation[9,10]. The multipotent ISCs give rise to

[1]National Institute of Biological Sciences, No. 7 Science Park Road, Zhongguancun Life Science Park, Beijing 102206, China. [2]Tsinghua Institute of Multidisciplinary Biomedical Research, Tsinghua University, Beijing 102206, China. [3]Present address: School of Life Science and Technology, ShanghaiTech University, Shanghai 201210, China. ✉e-mail: wangchh5@shanghaitech.edu.cn; xirongwen@nibs.ac.cn

committed progenitor cells including enteroblasts (EBs) and enteroendocrine progenitor cells (EEPs), which further differentiate into absorptive enterocytes (ECs) and secretory enteroendocrine cells (EEs), respectively (Fig. 1a)[11,12]. The ISC-EB interaction via Delta-Notch signaling guides EBs to differentiate into ECs, which occurs in approximately 80% ISC progeny[13]. A transient induction of Scute allows ISCs to generate EEPs, which divide once before terminal differentiation into pairs of EEs[14]. The differentiation of EE requires the expression of Prospero (Pros), a selector TF that governs and promotes the entire EE-specific transcriptional program[15]. Genetic manipulation or environmental stimulation has revealed the ability of lineage-committed progenitors to display lineage plasticity. For

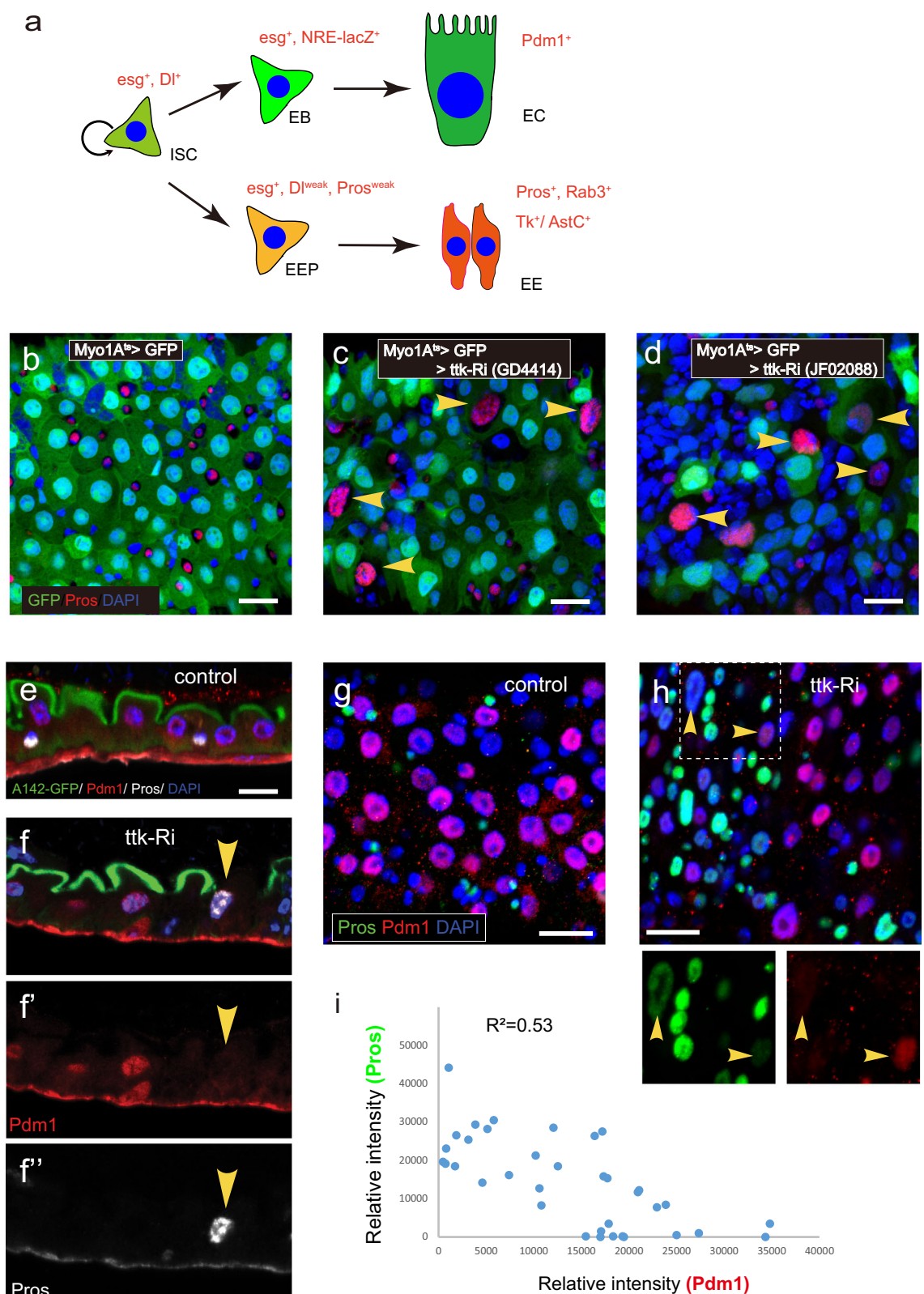

**Fig. 1 | Depleting Ttk in ECs causes transdifferentiation into EE-like cells. a** The diagram depicts the different cell types within the intestinal stem cell (ISC) lineage in the Drosophila intestine, along with cell type marker genes. **b–d** In normal guts, ECs are marked by Myo1A>GFP, while Pros is expressed in GFP-negative diploid cells (**b**). However, when Ttk is knocked down in ECs, Pros expression is activated in polyploid cells, concomitant with downregulation of GFP expression in these cells (**c**, **d**). **e**, **f** Sagittal-sectional views of the gut show that normal ECs express the signature transcription factor Pdm1 and can be labeled with A142-GFP at their apical brush border (**e**). However, upon Ttk knockdown in ECs, Pdm1 and A142-GFP expression is lost in polyploid cells, which instead show upregulation of Pros expression (**f**). **g–i** In normal guts, Pros and Pdm1 specifically mark EEs and ECs, respectively (**g**). Following Ttk depletion in ECs, the acquisition of Pros expression is accompanied by the loss of Pdm1 expression (**h**). Dot plots depicting the signal intensity of Pdm1 and Pros within each cell also reveal a significant negative correlation between these two factors (**i**), source data are provided as a Source Data file. Scale bars, 20 μm.

instance, EBs committed to the EC lineage can transition into the EE lineage by downregulating the fate-regulating TFs. Moreover, these EBs can undergo dedifferentiation into ISCs as part of the regenerative response triggered by pathogen infection[16–20]. Interestingly, EEs have multiple subtypes, and these subtypes can be efficiently switched from one type to another by temporarily altering their TF code[15,21]. In addition, the ectopic expression of the ISC factor *escargot (esg)* in ECs can suppress EC-specific differentiation gene expression[22]. Together, these observations suggest that ECs and EEs, despite being terminally differentiated cells, may still retain some degree of cell plasticity.

Previously, we discovered a transcriptional repressor known as Tramtrack (Ttk, or Ttk69 isoform) that functions to suppress EE specification in progenitor cells[16]. Depleting Ttk results in all progenitor cells adopting EE specification, regardless of the Notch activation status[16]. Our findings suggest that Ttk acts as a critical master repressor of EE specification during the early stages of progenitor cell differentiation. Interestingly, although Ttk plays a critical role in controlling cell fate specification in both ISCs and progenitor cells, we have found that it is most highly expressed in ECs[16]. This observation raises an intriguing possibility that Ttk may also play a role in ECs.

In this study, we identified an important role of Ttk in ECs, which is to maintain EC identity and prevent the transdifferentiation into EEs. Additionally, we discovered that EEs are also highly plastic and can dedifferentiate into ISCs when Pros are depleted. Our findings from chromatin accessibility analysis, as well as from a genetic screen aimed at identifying modifiers of EC-to-EE transdifferentiation have provided important insights into the molecular barriers that impede the process of cell reprogramming.

## Results

### Depleting Ttk in ECs initiates EC-to-EE transdifferentiation

To investigate the potential role of Ttk in ECs, we used an EC-specific GAL4 driver Myo1A-GAL4[ts] to express ttk-RNAi (VDRC#GD4414 or BDSC#TRiP.JF02088) and deplete Ttk in ECs for seven days. Strikingly, we observed that depleting Ttk caused approximately $19.5 \pm 2.1\%$ ($n = 17$, totally 1419 ECs counted) of differentiated ECs within the posterior midgut to express the EE signature protein Pros (Fig. 1b–d; Supplementary Fig. 1a, b). The acquired EE-like identity in Ttk-depleted ECs was accompanied by the loss of EC identity, which was reflected by reductions in EC-specific markers including Myo1A-GAL4, UAS-GFP (or Myo1A>GFP hereafter, for simplicity) (Fig. 1b–d), the brush border marker A142-GFP (Fig. 1e, f), and the EC-specific TF Pdm1 (Fig. 1g, h). Collectively, these data suggest that the acquisition of EE identity in Ttk-depleted EC is accompanied by the loss of EC identity.

To determine if the transdifferentiation of ECs to EEs occurred directly through a switch in transcriptional programs or indirectly through dedifferentiation into a progenitor cell state before adopting a different cell fate, we analyzed the expression dynamics of Pdm1 and Pros following Ttk depletion in ECs. We observed that both the expression levels of the EC-specific TF Pdm1 and the EE-specific TF Pros exhibited an inverse correlation in each individual Ttk-depleted EC. Occasionally some ECs expressed both Pdm1 and Pros at low levels (Fig. 1i, arrowheads), and in general, the appearance of Pros expression seemed to occur immediately after the loss Pdm1 (Fig. 1g–i). Furthermore, we conducted staining for the progenitor cell marker esg-lacZ

during the reprogramming process. However, no discernible signal was observed in ECs depleted of Ttk (Supplementary Fig. 2). These observations indicate that the transdifferentiation is direct and not through a progenitor cell state.

To address the concern that late EBs might already exhibit Myo1A-GAL4 expression, potentially leading to a shift in the differentiation pathway instead of transdifferentiation in differentiated ECs, we conducted co-staining experiments using the EB-specific marker NRE-lacZ along with Myo1A>GFP. Interestingly, we observed co-localization of lacZ in several GFP+ cells with smaller nuclear size (Supplementary Fig. 3a, indicated by arrowheads), suggesting the presence of Myo1A-Gal4 expression in EBs. We therefore employed Mex1-GAL4, an alternative EC driver[23], to determine if Ttk depletion still induced transdifferentiation. Mex1-Gal4 displayed more specificity in ECs, as no colocalization was observed with the EB marker NRE-lacZ (Supplementary Fig. 3b). Importantly, knocking down Ttk using Mex1-Gal4 similarly activated Pros in differentiated cells, leading to the loss of Mex1>GFP expression in these transformed cells (Supplementary Fig. 4a, b, indicated by arrowheads).

To further exclude the possibility of an altered differentiation pathway during the EB stage leading to the emergence of Pros+ polyploid cells (cells with a nuclear size greater than 20 μm²), we employed NRE-GAL4 to deplete Ttk specifically in EBs. Remarkably, this approach resulted in the transdifferentiation of EBs into entirely diploid Pros+ cells, with no presence of polyploid Pros+ cells observed (Supplementary Fig. 3d, e). Taken together, these findings provide compelling evidence supporting the conclusion that Ttk depletion has the capacity to directly induce transdifferentiation of cell fate from ECs to EE-like cells.

### The transdifferentiated cells exhibit functional EE characteristics

EEs are responsible for regulating various physiological processes by secreting a range of peptide hormones[24]. In *Drosophila*, class I and II EE subtypes secrete Tachykinin (Tk) and Allatostatin C (AstC), respectively[21]. We found that a small subset of the Pros+ ECs that underwent EC-to-EE transdifferentiation expressed Tk, AstC, and Rab3, suggesting that these cells have acquired the hormone-producing function of EEs (Fig. 2a–f and Supplementary Figs. 1c, d, 4c–f). Interestingly, these transdifferentiated cells exhibit a similar pattern to normal EEs in terms of mutually exclusive expression of Tk and AstC, indicating that the subtype feature of EEs is maintained in the transdifferentiated cells (Supplementary Fig. 5a, b). To assess the stability of the transformed cell population induced by Myo1A>ttk-RNAi, we subjected the flies back to the permissive temperature of 18 °C for a duration of 7 days. Interestingly, during this period, the Tk+ polyploid cells persisted, indicating the presence of a stable transformed cell population rather than a transient activation of Ttk-target gene expression (Supplementary Fig. 5c).

Additionally, our electron microscopy analysis showed that the putative transdifferentiated cells contained numerous secretory granules (Fig. 2h, arrowhead). This characteristic is typically associated with secretory cells, whereas normal ECs are characterized by the presence of lipid droplets (Fig. 2g, arrow). Taken together, these observations suggest that the depletion of Ttk leads to a conversion of ECs into cells that molecularly and functionally resemble EEs.

 

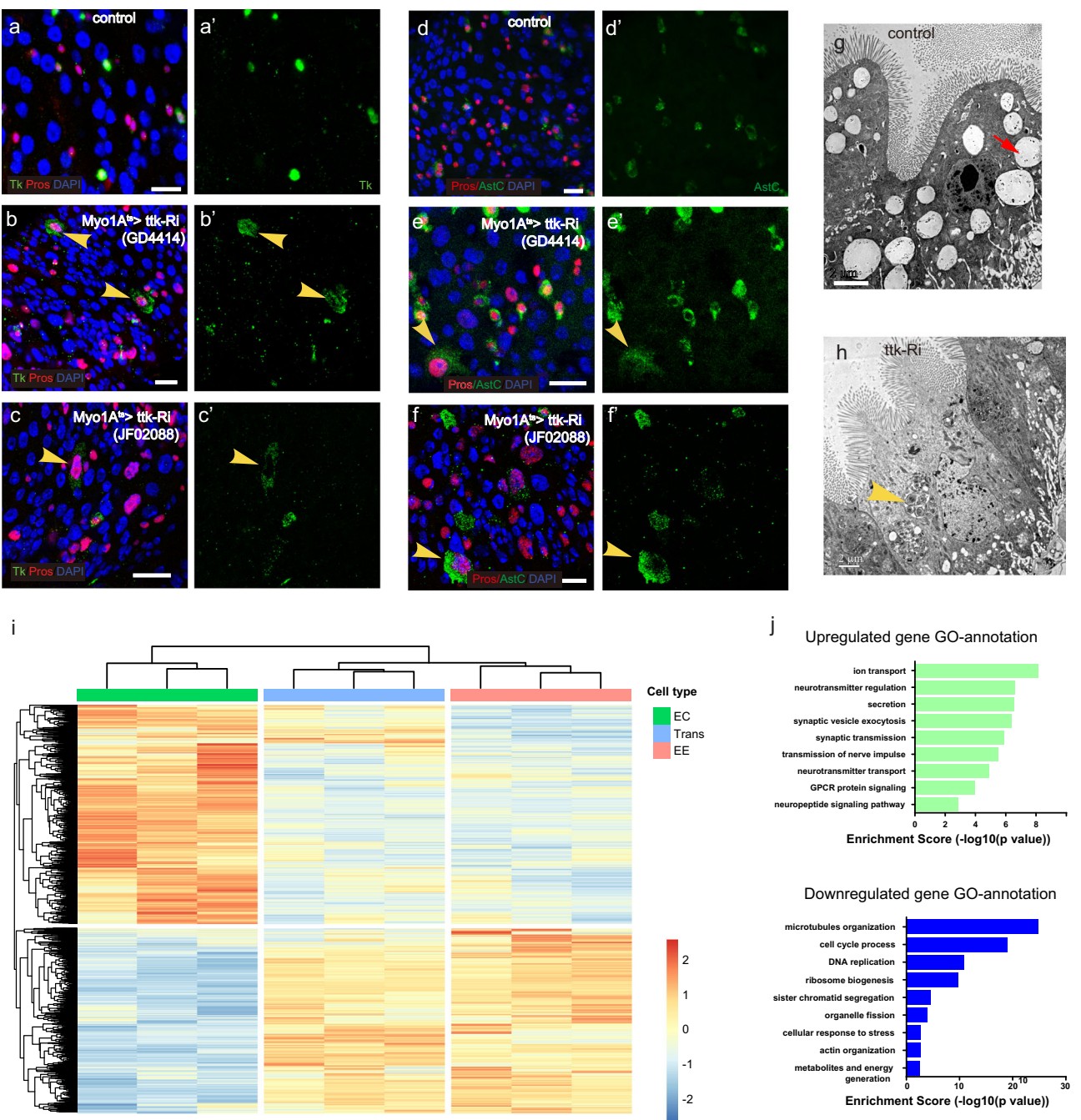

**Fig. 2 | The transdifferentiated cells exhibit functional EE characteristics. a–f** In normal guts, peptide hormones Tk and AstC are exclusively expressed in diploid EEs (**a**, **d**). However, upon Ttk knockdown in ECs, Tk and AstC expression are observed in polyploid cells (**b**, **c**, and **e**, **f**, indicated by yellow arrowheads). **g**, **h** Transmission electron microscopy (TEM) images of normal and ttk-depleted guts. Abundant lipid droplets are present in normal ECs (g, red arrow), while they are diminished in Ttk-depleted ECs, with small secretory granules appearing instead (**h**, yellow arrow). **i** Transcriptome profiles of normal ECs, EEs, and Ttk-depletion-induced EE-like cells. The transdifferentiated EE-like cells, induced by Ttk depletion in ECs, exhibit a general shift in their transcriptome towards an EE-like state. **j** Gene ontology analysis of upregulated (left panel) and downregulated genes (right panel) upon Ttk depletion in ECs. Statistical test of *p* value: modified Fisher's exact test (EASE score). Scale bars, 20 µm.

## Genome-wide RNA-seq analysis reveals a complete EC-to-EE transcriptome switch in Tk⁺ polyploid cells

To determine to what extent transdifferentiation occurs in order for hormone-producing function to be obtained, we sorted TK⁺ polyploid cells using TK^En^-GFP, a GFP reporter driven by the TK gene enhancer[21], and performed RNA sequencing analysis.

We found that EC-derived TK⁺ polyploid cells had virtually identical gene expression patterns to EEs, with highly expressed programs such as neurotransmitter signaling, vesicle secretion, and neuropeptide signaling (Fig. 2g, h left panel, and Supplementary Data 1). Conversely, many EC-specific programs, including cytoskeleton organization, DNA replication, metabolic and energy production, and responses to stress, were downregulated in TK⁺ ECs (Fig. 2h, right panel). The analysis of a comprehensive collection of gene sets associated with EC and EE functions further validated the observed transcriptome shifts (Supplementary Fig. 1e, f). Based on these

observations, we propose that, following Ttk ablation in ECs, a small subset of Pros[+] cells undergoes a complete switch in cell identity from ECs to EEs in terms of both gene expression and function, despite the cells still remaining polyploid. We therefore speculate that the acquisition of hormone-producing function of EEs may require a complete transcriptional reprogramming, that is, a genome-wide transcriptome shift from that of EC to EE.

## Pros mediates EC-to-EE transdifferentiation

Ttk suppresses the specification of EE fate by repressing *achaete-scute* genes, whose expression promotes EE fate specification by inducing Pros expression[16]. We recently showed that Pros acts as a selector factor for EE fate[15], and this may explain why transdifferentiation towards EE occurs following Ttk depletion in ECs. We found that overexpression of Ttk in EEs was able to eliminate Pros expression in EEs (Fig. 3a, b), supporting a role for Ttk in suppressing Pros expression. Knockdown of Pros completely abolished Ttk depletion-induced EC-to-EE transdifferentiation, and the expression of EC-specific markers (Myo1A>GFP and Pdm1) failed to be downregulated (Fig. 3c, e–h), with no activation of peptide hormones observed (Fig. 3d). Therefore, Pros is a key mediator for Ttk depletion-induced EC-to-EE transdifferentiation.

We previously showed that the ectopic expression of Pros in ECs can induce AstC expression in some ECs, indicating that Pros may be sufficient to induce EC-to-EE transdifferentiation[15]. In fact, overexpression of Pros robustly suppressed the expression of EC marker genes, such as Myo1A>GFP and Pdm1, indicating a cell identity switch (Fig. 3g, h). To analyze the role of Pros in EC-to-EE transdifferentiation systemically, we isolated ECs from Pros overexpressed guts (Myo1A[ts]>pros), and Ttk & Pros co-depleted guts (Myo1A[ts]>ttk-RNAi & pros-RNAi), respectively, and performed transcriptome analysis by RNA sequencing. By comparing them to the transcriptome of normal EEs and ECs, we found that overexpression of Pros alone in ECs was able to globally upregulate EE-specfic genes and downregulate a significant portion of EC-specific genes (Fig. 3i, j and Supplementary Data 2), although the expression levels of the up- or down-regulated genes were not comparable to the levels in normal EEs or ECs, likely due to incomplete or non-occurring transdifferentiation in many ECs that were sorted out for the analysis. Importantly, depletion of Pros virtually prevented Ttk depletion-induced transcriptome switch from EC to EE (Fig. 3i). These observations demonstrate that Pros is the key downstream factor that mediates EC-to-EE transdifferentiation.

The ability of Pros to suppress EC-specific markers suggests that this selector protein not only activates EE-specific transcriptional programs but also inhibits EC-specific transcriptional programs. This dual function may underlie its ability to promote EC-to-EE transdifferentiation.

## Depletion of pros in EEs causes cell dedifferentiation into intestinal progenitor cells

Given the high degree of cell plasticity observed in differentiated ECs, we investigated whether differentiated EEs possess similar properties. Our recent study showed that depleting Pros in EEs leads to a nearly complete loss of EE-specific transcriptional programs[15]. Here we revisited the transcriptome data of Pros-depleted EEs and compared it with that of normal EEs. We found that, apart from the 1051 down-regulated genes that belong to the EE signature gene sets, approximately 1700 genes were significantly upregulated (Supplementary Data 3). GO analysis of these upregulated genes indicated enrichment in genes involved in mitosis (Fig. 4a). This raised the possibility that these cells had acquired characteristics of intestinal progenitor cells, as EEs and ECs are typically post-mitotic cells. To test this possibility, we examined the expression of the top 250 progenitor cell-specific genes,

and the Gene Set Enrichment Analysis (GSEA) plot showed significant upregulation of this gene set in Pros-depleted EEs (Fig. 4b, Supplementary Data 4).

We further analyzed the transcription levels of the previously-defined top 32 progenitor cell-specific TFs[25], including *esg*, *sox100B*, *fkh*, and *klu*, and additional progenitor cell enriched genes including cell cycle regulators, and cell polarity and cell adhesion-related genes, and found that all of these genes were upregulated to some extent following Pros depletion (Fig. 4c). Immunostaining of the gut confirmed that the expression of *esg* was activated in about 36.5% ± 5.6% Pros-depleted EEs (Fig. 4d, e, j, and Supplementary Fig. 6a). Similarly, the expression of Sox100B, which is typically present in ISCs and EBs[26], was activated in Pros-depleted EEs (Fig. 4f, g). Interestingly, about 30.5% ± 3.7% of the ProsV1>GFP[+] cells regained Dl expression following Pros depletion, whereas Dl was rarely expressed in control ProsV1>GFP[+] cells (Fig. 4h, i, k). In addition, consistent with the derepression of cell cycle genes, the mitotic marker PH3 was detected in the Pros-depleted ProsV1>GFP[+] cells, suggesting that Pros depletion enables these EEs to re-enter mitotic cell cycle (Fig. 4l, m). Collectively, these observations indicate that Pros-depleted EEs can become ISCs by undergoing dedifferentiation.

## The dedifferentiated cells can divide and differentiate into ECs

We next performed cell lineage tracing of Pros-depleted EEs using the UAS-flp and flp-out cassette system to test our hypothesis. This system allows for labeling cells of interest and their progeny over time. On day 3 after inducing lineage labeling, we found that in control samples, only Pros[+] cells, but not Dl[+] were present in the lineage, and it was extremely rare to see cells in the lineage not being Pros[+] (Fig. 4n). In contrast, many diploid cells in the lineage of Pros-depleted EEs showed Dl expression, suggesting acquisition of ISC identity. In addition, there were apparently more lineage-labeled cells in the EE>pros-RNAi gut epithelium, indicating cell proliferation may have occurred (Fig. 4o, yellow arrow). On day 7 after inducing lineage labeling, we found that again in control samples, only Pros[+] cells, not Dl[+] cells, were present in the labeled lineage (Fig. 4p, white arrow). In contrast, lineage cells of Pros-depleted EEs became diversified, with some remaining diploid with or without Dl expression, and others differentiating into ECs as evidenced by polyploid nuclei and Pdm1 expression (Fig. 4q–r, green arrow).

To address the concern that dedifferentiation might occur in early progenitor EEPs rather than in differentiated EEs, we employed an alternative EE-specific Gal4 driver named Tk-GAL4, which is expressed specifically in class II EE subtypes, and performed cell lineage tracing experiments to track the fate of Tk[+] EEs after depleting Pros. Interestingly, we observed the emergence of Dl[+] diploid cells and Pdm1[+] polyploid cells within the Tk lineage cells, while lineage-negative EEs retained their Pros[+] cell identity (Fig. 4t–u). These findings provide further evidence that the dedifferentiation process occurs in differentiated EEs rather than in early progenitor EEPs.

These results demonstrate that the depletion of Pros in differentiated EEs triggers robust cell dedifferentiation leading to the acquisition of ISC identity. These findings highlight a dual function of Pros in EEs. One is to act as a selector protein to establish and maintain EE-specific transcriptional programs. Another is to concurrently suppress stem cell-specific transcriptional programs. Indeed, transient overexpression of Pros in ISCs promotes their exit from the cell cycle and differentiation into EEs[16]. Moreover, Pros is known to have a dual role in the neuroblast-neuron lineage. It directly suppresses the transcription of cell cycle genes and neuroblast-specific genes, thereby inhibiting neuroblast self-renewal. At the same time, it directly promotes the transcription of neuron-related genes, leading to neuron specification[27,28]. This dual function of Pros may explain the EE dedifferentiation phenotype upon depletion of Pros.

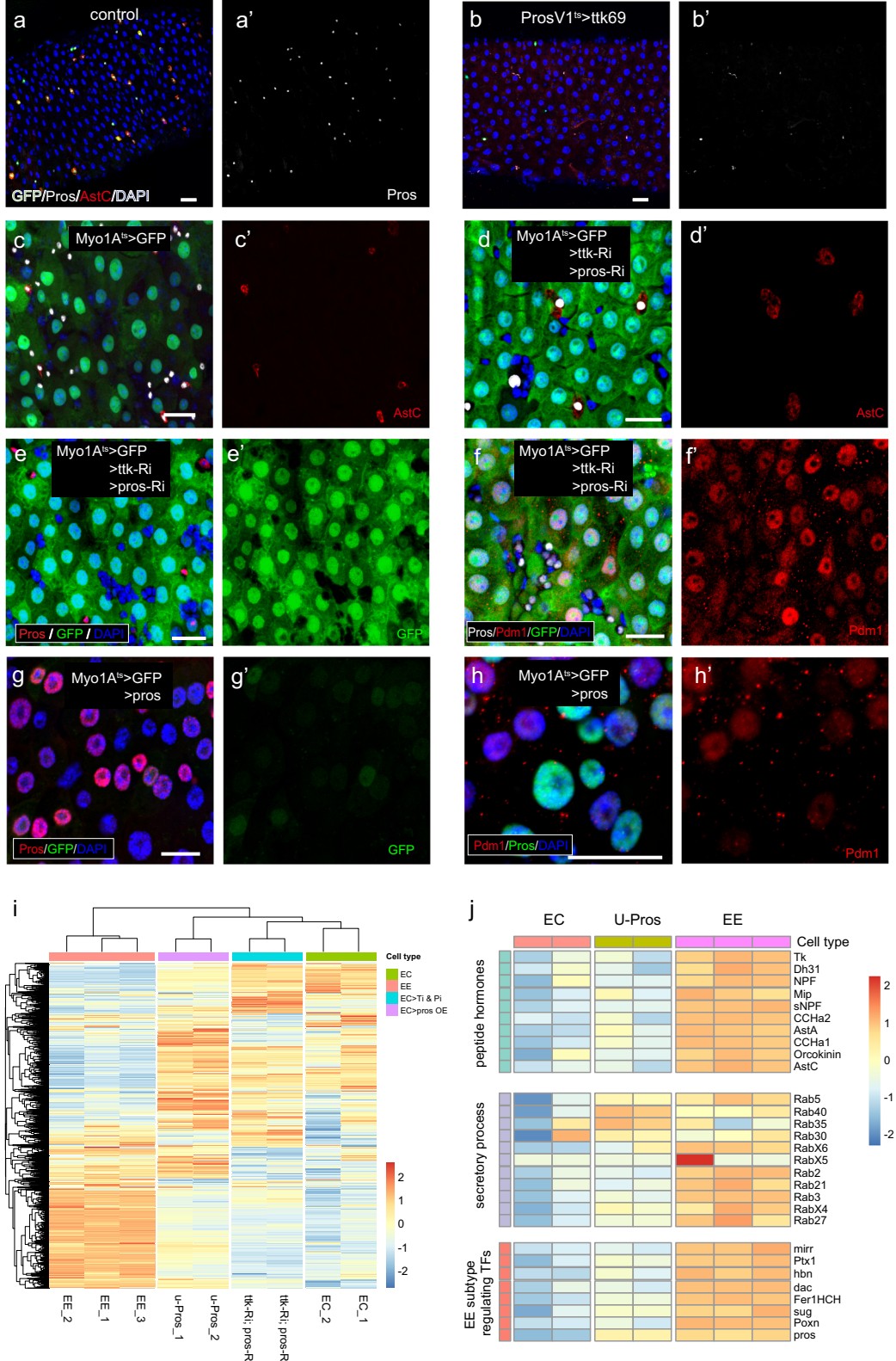

**Fig. 3 | Ttk suppresses pros to prevent EC-to-EE transdifferentiation.**
**a**, **b** Overexpression of Ttk69 in EEs leads to the suppression of Pros expression and loss of peptide hormones. **c**, **d** Depleting Pros expression in Ttk-depleted ECs using pros-RNAi prevents the expression of AstC. **e**–**h** Depleting Pros expression in Ttk-depleted ECs using pros-RNAi abolishes the suppression of Myo1A-GFP (**e**) and Pdm1 expression (**f**). Conversely, overexpressing Pros alone is sufficient to suppress Myo1A-GFP (**g**) and Pdm1 expression (**h**) in ECs. **i** A heatmap displays the expression patterns of differentially expressed genes between ECs and EEs in EC>Pros-OE and EC>ttk-RNAi & pros-RNAi cells. A general transcriptome reprogramming towards an EE identity is observed in EC>Pros-OE cells, while co-depleting Pros prevents the expression of EE identity genes induced by Ttk-RNAi. **j** Another heatmap illustrates the expression of selected EE signature genes in normal EEs, ECs, and EC>Pros-OE cells. An overall activation of EE identity genes, including peptide hormones, secretory factors, and EE subtype-regulating TFs, is observed in ECs overexpressing Pros. Scale bars, 20 µm.

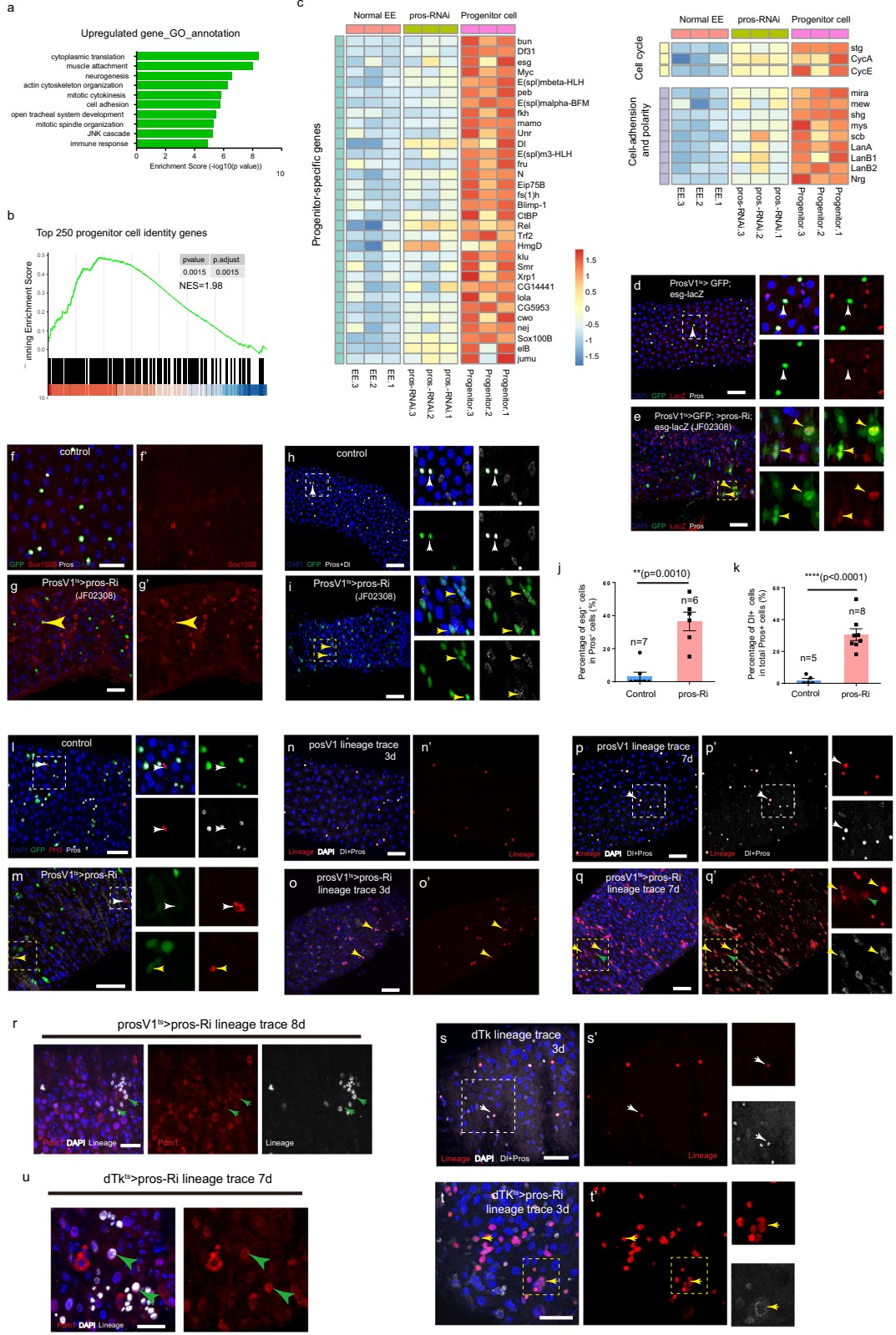

## A shared chromatin accessibility profile among EC, EE, esg+ progenitor cells and transdifferentiated cells

The above observations of transdifferentiation or dedifferentiation suggest high cell plasticity for both types of differentiated intestinal cells. In mammalian ISC lineage, secretory progenitor cells can differentiate into enterocytes upon *atoh1* depletion, and chromatin accessibility analysis shows that both secretory and absorptive progenitor cells share a genome-wide similarity in their profiles, rendering their transdifferentiation potential[29]. We wondered whether ECs and EEs in the *Drosophila* midgut, despite being terminally differentiated cells, shared a similar chromatin accessibility profile.

To explore this, we performed ATAC-seq (Assay for Transposase-Accessible Chromatin using sequencing) analysis of FACS-sorted esg+ progenitor cells, EEs, and ECs. We first compared our ATAC-seq data to

**Fig. 4 | EE-to-ISC dedifferentiation following the depletion of pros in EEs. a** GO analysis of genes upregulated upon Pros depletion in EEs. Statistical test of *p* value: modified Fisher's exact test (EASE score). **b** GSEA plot of the top 250 progenitor identity genes in transcriptome alterations induced by Pros depletion. Statistical test of *p* value: empirical phenotype-based permutation test. **c** Heatmap displaying the expression of a set of progenitor signature genes in progenitor cells, control EEs, and Pros-depleted EEs. **d, e** Pros depletion leads to the expression of esg in prosV1-GAL4 > GFP$^+$ cells outside of the R3 region (yellow arrows in **e**), while rarely detected in the normal gut (white arrows in **d**). **f, g** Sox100B is not detected in normal EEs (**f**), but is ectopically activated in Pros-depleted EEs (**g**, yellow arrow). **h, i** Pros depletion leads to the activation of Dl in prosV1-GAL4 > GFP$^+$ cells (yellow arrows in **i**), while rarely detected in GFP$^+$ EEs of the normal gut (white arrows in **k**). **j, k** Quantification of the percentage of esg+ (**j**) or Dl+ (**k**) cells in GFP$^+$ EEs outsize

R3. Error bars represent Mean ± SEM, \*\**p* < 0.01; \*\*\*\**p* < 0.0001 (two-tailed Student's *t* test); "n" indicate the number of guts used for quantification; source data are provided as a Source Data file. **l, m** Pros depletion leads to the detection of mitotic PH3 signal in prosV1-GAL4 > GFP$^+$ cells (yellow arrows in **m**), in addition to GFP$^-$ PH3$^+$ cells (white arrows in **l** and **m**). **n–r** Lineage tracing of normal and Pros-depleted EEs after tracing for 3 days (**n–o**) and 8 days (**p–r**). Knocking down Pros induces Dl expression in many lineage-labeled cells (**o, q**, yellow arrow). Pdm1$^+$ polyploid cells can be found in the lineage-labeled cells after tracing for 8 days (q, r, green arrow). (s-u) Lineage tracing of normal (s) and Pros-depleted Tk$^+$ EEs(t, u) using Tk-GAL4 driver. After tracing for 3 days, Pros depletion leads to lineage+ small cell clusters, and some of them regained Dl expression (t, yellow arrow). After tracing for 7 days, Pdm1$^+$ polyploid cells could be observed in lineage-labeled cells (u, green arrow). Scale bars, 50 μm.

previously published ATAC-seq data on ISCs[30], as well as Elav$^+$ larval neurons, germline cells and eye imaginal disc cells[31–33]. As shown in a PCA plot, cells from different lineages were all distantly separated (Fig. 5a), which is consistent with the idea that lineage-restricted genes become separately accessible during embryo development from a fertilized egg to specify distinct germ layers and organs[34]. Interestingly, all intestinal cells, including progenitor cells and terminally differentiated cells, were clustered together in the PCA plot (Fig. 5a). Despite ISCs, ECs and EEs having very different transcriptome profiles, with the Pearson correlation scores ranging from 0.6 to 0.84 (Fig. 5b), their chromatin accessibility profiles showed higher similarity, with the correlation scores ranging from 0.78 to 0.88 (Fig. 5c).

To investigate the chromatin loci directly relevant to cell type-specific physiological functions, we selected the top 1000 transcribed genes specific to EC, EE, and progenitor cell and analyzed the chromatin accessibility in these gene loci (Fig. 5d). Again, despite significant differences in transcriptional activity, these gene loci exhibited strikingly similar ATAC-seq profiles across all the cell types analyzed.

We further investigated the chromatin loci of *esg*, *pros*, and *nub*, which are representative TFs for progenitor cells, EEs, and ECs, respectively. *esg* is specifically expressed in intestinal progenitor cells and is essential in preventing cell differentiation[22,35]. As previously mentioned, *pros* is EE-specific and a selector gene for the establishment and maintenance of EE identity[15], while *nub* is specifically expressed in ECs and regulates EC differentiation[22]. Despite pronounced differences in their transcriptional activities among the different cell types, all three genes exhibited open chromatin status in all three cell types, although the *pros* locus manifested unique accessible chromatin in EEs (Fig. 5e).

The observed similarity in chromatin accessibility among intestinal lineages may account for the cell fate plasticity observed among different intestinal cell types. It is therefore conceivable that depletion of cell identity maintenance TFs leads to transcriptome shifts without affecting the original chromatin accessibility landscape. To confirm this, we compared the chromatin accessibility profiles of the differentially-expressed genes in normal EEs and Pros-depletion-induced progenitor-like cells, or normal ECs and Ttk-depletion-induced EE-like cells. The Pearson correlation scores revealed a significant positive correlation in the ATAC-seq profiles between cell groups before and after Pros (0.89) or Ttk (0.65) depletion. These data indicate that despite the global changes in the transcriptome, the chromatin accessibility profile remained largely unchanged following the cell fate switches (Fig. 5f, g). This suggests that the underlying chromatin landscape remains relatively stable during the process of cell fate reprogramming.

## Age factor as a potential barrier to the EC-to-EE transdifferentiation

The discovery of EC-to-EE transdifferentiation following Ttk depletion in *Drosophila* provides an in vivo genetic system to understand the molecular mechanisms underlying cell plasticity. As mentioned earlier,

about 20% of polyploid cells successfully turns on Pros expression following Ttk depletion, and the percentage of cells successfully turning on peptide hormone expression is further limited, indicating heterogeneity in cell plasticity among ECs.

One potential explanation for the variation in cell plasticity among enterocytes (ECs) is that newly formed ECs exhibit high plasticity, which gradually decreases with age (or with increased maturation or polyploidy). To investigate this hypothesis, we can employ an approach that involves identifying specific markers for young/ newly-formed and aged/pre-existing ECs, and then comparing the efficiency of EC-to-EE transdifferentiation following Ttk depletion in these different age groups. The TF Sox21a has been shown to play a critical role in promoting EC differentiation[36,37]. It is expressed at low levels in progenitor cells but is highly upregulated in differentiating and early ECs[36]. Therefore, Sox21a could serve as a marker to distinguish between young and aged ECs.

Given that the turnover rate of the intestinal epithelium is typically slow under normal conditions, we induced epithelial renewal by subjecting the flies to heat shock treatment, so that more differentiating ECs with Sox21a expression could be observed. We then examined Sox21a expression in two known EC-specific GAL4 lines, Myo1A-GAL4 and mex1-GAL4. In the unstressed gut, Myo1A-GAL4 and mex1-GAL4 showed largely similar expression patterns, except that the expression level of mex1>GFP was more variable among ECs (Fig. 6a, b). In the stressed guts, however, many Myo1A-GAL4 > GFP$^+$ cells exhibited high levels of Sox21a expression, whereas only a few Mex1-GAL4 > GFP$^+$ cells displayed high Sox21a expression, indicating Myo1A-GAL4 > GFP, but not Mex1>GFP frequently marks early-differentiating ECs (Fig. 6d, e, arrowheads). Additionally, we screened several enhancer trap GAL4 lines from different sources and identified GMR23G10-GAL4 as another EC-specific driver. Interestingly, in the unstressed gut, GMR23G10-GAL4 was expressed only in a subset of ECs in the posterior midgut, displaying a mosaic expression pattern (Fig. 6c). Co-staining with anti-Sox21a in the stressed gut revealed that GMR23G10-GAL4 was exclusively expressed in Sox21a$^-$ ECs, and many polyploid cells that still retained low levels of Sox21a, presumably the newly differentiated ECs or young ECs, are negative for GMR23G10 > GFP expression. This observation indicates that GMR23G10-GAL4 marks elder ECs only (Fig. 6f). Based on these observations, we consider Myo1A-GAL4 as a driver for both young and aged ECs, mex1-GAL4 for aged/ pre-existing ECs predominantly, and GMR23G10-GAL4 for aged/ pre-existing ECs specifically.

We next analyzed and compared the EC-to-EE transdifferentiation efficiency in these different EC populations. Knocking down Ttk in Mex1-GAL4 > GFP$^+$ ECs caused approximately 14.1% ± 1.3% (60/426) of the cells to turn on Pros expression, and only a small number of cells (0.58 ± 0.19 cells per gut, *n* = 12 guts) showed Tk expression, suggesting that these cells have a lower transdifferentiation efficiency than the Myo1A>GFP$^+$ cells (Fig. 6g, i, j). Interestingly, knocking down Ttk in GMR23G10 > GFP$^+$ ECs caused approximately 8.9 ± 1.8% (60/727) of cells to turn on Pros expression (Fig. 6h, i), but none of these cells

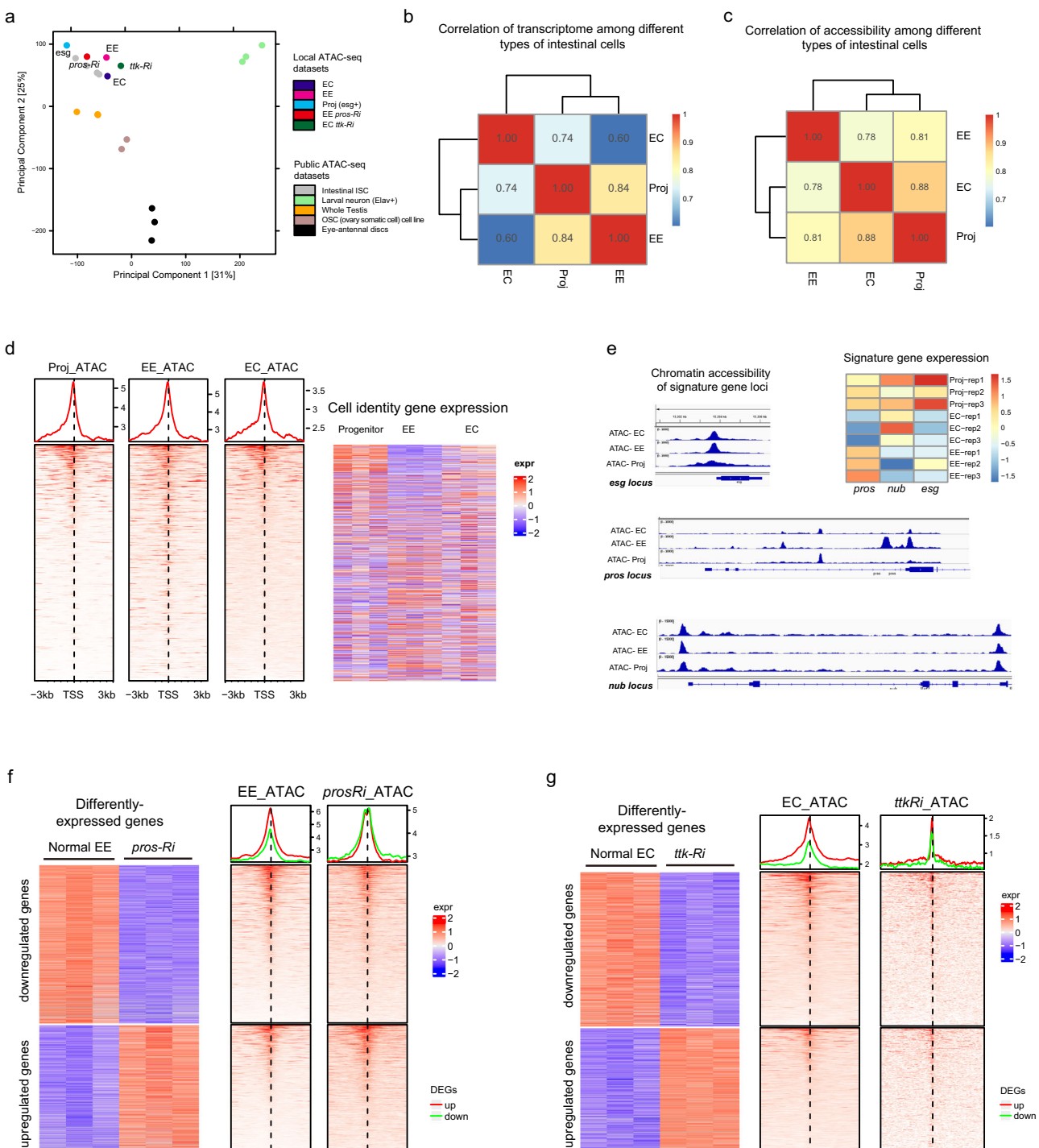

**Fig. 5 | Broadly similar chromatin accessibility landscape among different intestinal cell types. a** Principal Component Analysis (PCA) plot displaying the clustering of ATAC-seq profiles of intestinal epithelium cells, including profiles of ECs, EEs, progenitor cells, as well as pros-depleted EEs and Ttk-depleted ECs generated in this study, with published ATAC-seq profiles of intestinal cells and cells from other organs including ovary, testis, imaginal disc, and brain. **b** Pearson correlation coefficient analysis of RNA sequencing profiles in progenitor cells (esg⁺), ECs, and EEs. **c** Pearson correlation coefficient analysis of chromatin accessibility profiles (ATAC-seq profiles) in progenitor cells (esg⁺), ECs, and EEs. **d** Integrated map displaying a heatmap of expression levels of the top 1000 identity genes for progenitor cells, EEs, and ECs (right), along with the corresponding chromatin

accessibility profiles of these identity genes in esg⁺ progenitor cells, EEs, and ECs (left). **e** Chromatin accessibility profiles (left) and transcription levels (right) of Esg, Pros, and Pdm1, which are representative TFs of progenitor cells, EEs, and ECs, respectively. **f** Integrated map showing a heatmap of significantly altered genes between normal and pros-depleted EEs (left), along with the chromatin accessibility profiles of these differentially expressed genes in normal and pros-depleted EEs (right). **g** Integrated map displaying a heatmap of significantly altered genes between normal and Ttk-depleted ECs (left), along with the chromatin accessibility profiles of these differentially expressed genes in normal ECs and Ttk-depleted ECs (right).

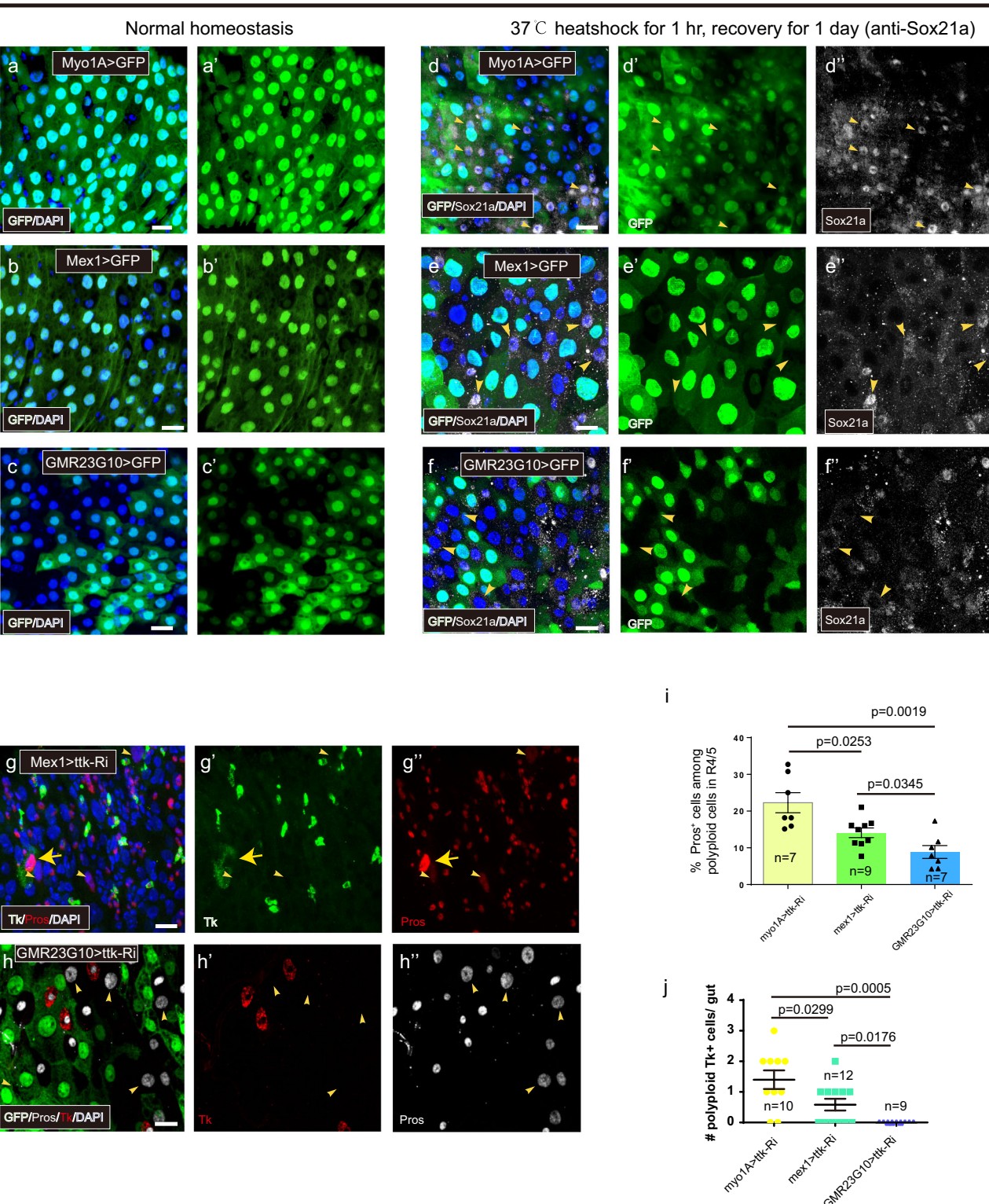

**Fig. 6 | Age factor as a potential barrier to the EC-to-EE transdifferentiation.**
**a**–**f** The expression patterns of myo1A>nGFP, Mex1>nGFP, and GMR23G10>nGFP in normal (**a**–**c**) and stressed (**d**–**f**) guts. The stressed gut is co-stained with anti-Sox21a (in white). Arrowheads indicate Sox21a^high (**a**, **b**) and Sox21a^low cells (**c**). **g** Depleting Ttk in ECs using mex1-GAL4 leads to Pros (arrowheads) or Tk (arrows) in a subset of polyploid cells. **h** Depleting Ttk in ECs using GMR23G10-GAL4 leads to Pros (arrowheads) but not Tk expression in a subset of polyploid cells. **i**, **j** Quantification of the percentage of Pros^+ (**i**) or the number of Tk^+ (**j**) polyploid cells in the posterior midgut upon Ttk-depletion driven by myo1A-GAL4; mex1-GAL4, and GMR23G10-GAL4, respectively; Error bars represent Mean ± SEM, *$p < 0.05$, **$p < 0.01$, ***$p < 0.001$ (two-tailed Student's $t$ test). "n" indicate the number of guts used for quantification; source data are provided as a Source Data file. Scale bars, 20 μm.

(totally 727 cells examined) showed Tk expression (Fig. 6h, j), indicating that GMR23G10-GAL4 > GFP[+] ECs have the lowest transdifferentiation efficiency. These data support the idea that cell plasticity is high in young or newly differentiated ECs and declines rapidly with maturation/ age.

The process of EC differentiation, maturation, and aging is often accompanied by an increase in endoreplication, which involves DNA replication without cell division, leading to polyploidy[38,39]. This raises the question of whether polyploidy can hinder cell transdifferentiation. To investigate this, we employed RNAi to knock down E2F1, a TF essential for driving EC endoreplication. As a result, we observed a significant reduction in nuclear sizes of myo1A>GFP[+] cells (Supplementary Fig. 7a, b), indicating successful interference with the endoreplication process. Interestingly, when E2F1 was co-depleted with Ttk, we observed a mild but significant increase in Pros[+] cells within the myo1A>GFP[+] cell population (Supplementary Fig. 7c, d). This finding suggests that polyploidy might act as a barrier during the process of EC-to-EE transdifferentiation.

### Lineage confliction as a barrier to the EC-to-EE transdifferentiation

Conflicting alternative lineages have been indicated as potential barriers against reprogramming in mammalian studies[40,41], and our previous work showed that Ttk depletion in ISCs caused the derepression of several neuroblast-specific TFs, including Deadpan (Dpn), which is a neural stem cell factor that, when activated in ISCs, drives self-propagation and prevents EE differentiation[42]. We speculated that if Ttk depletion in ECs led to the derepression of neuroblast genes, lineage conflict could potentially occur and prevent EC-to-EE transdifferentiation. Although the RNA-seq data of the transdifferentiated cells did not capture any expression of the neuroblast genes, we detected Dpn protein expression, by surprise, in many ECs in Myo1A>ttk-RNAi guts. By co-staining with Pros, we found that the expression of Dpn and Pros in ECs were mutually exclusive (Supplementary Fig. 8a, b).

To further examine the inverse relationship between Dpn and Pros expression, we generated a pros-mCherry knock-in line which yields a Pros Protein fused with mCherry at its C-terminal, and used it to sort out Pros-activated and Pros-not-activated polyploid cells in Myo1A[ts]>ttk-RNAi guts and performed quantitative PCR analysis. There was indeed a significant upregulation of Dpn expression in Pros-not-activated cells compared to that in Pros-activated cells (Supplementary Fig. 8c).

To investigate whether Dpn depression acts as a barrier to EC-to-EE transdifferentiation, we simultaneously depleted Dpn and Ttk in ECs, and observed significantly increased percentage of polyploid cells turning on Pros, Tk or AstC: approximately $65.3 \pm 7.1\%$ ($n = 5$ guts) of polyploid cells turned on Pros expression, and on average $8.6 \pm 1.7$ ($n = 7$ guts) polyploid cells turned on AstC expression (Supplementary Fig. 8d–g), showing approximately 3–5 fold increase in EC-to-EE transdifferentiation efficiency.

These observations suggest that the derepression of neural stem cell factors and EE-promoting factors following Ttk depletion can cause a lineage-conflict scenario in which Dpn exerts a negative effect on Pros activation, thereby preventing EC-to-EE transdifferentiation.

### A genetic screen identified NuRD complex as a barrier to the EC-to-EE transdifferentiation

The establishment of this in vivo EC-to-EE transdifferentiation system in *Drosophila* provided an opportunity to identify new regulators involved in reprogramming through genetic screens. In this regard, we screened about 120 chromatin-related regulators, including polycomb group genes, regulators for heterochromatin formation, chromatin remodeling factors, and histone modification enzymes (Supplementary Data 5), and aimed to identify those whose depletion could

increase transdifferentiation efficiency following Ttk depletion (Fig. 7a). From this screen, we identified MEP-1 and Mi-2, both encode essential components of the NuRD nucleosome remodeling complex[43,44]. We found that simultaneous knockdown of MEP-1 and Ttk (Myo1A[ts]>MEP-1-RNAi & ttk-RNAi) or dMi-2 and Ttk in ECs (Myo1A[ts]> dMi-2-RNAi & ttk-RNAi) led to a significant increase in Pros[+] ECs (approximately $85.6 \pm 1.8\%$ and $81.3 \pm 2.04\%$ ECs turned on Pros, respectively) or TK[+] ECs ($5.7 \pm 0.4$ and $5.8 \pm 0.5$ ECs per R5 region turned on Tk, respectively) (Fig. 7b–g), indicating enhanced transdifferentiation. As controls, knockdown of MEP-1 or dMi-2 alone did not produce any obvious phenotype (Supplementary Fig. 9). As the NuRD complex possesses histone deacetylase activity for the establishment of local repressive transcription, and has been implicated in Ttk and other transcriptional factor-mediated transcriptional repression[44,45], it may provide an epigenetic memory of local repressive transcriptional status in ECs, thereby acting as a barrier to transdifferentiation. Interestingly, NuRD has been implicated as an important regulator of reprogramming in mammalian cells[46].

## Discussion

Our study highlights a paradigm of in vivo cell reprogramming through the depletion of a single endogenous factor. Although in vivo transdifferentiation has been observed in some cases across various species[2], it is often rare, inefficient, or not amenable to genetic screens. The in vivo transdifferentiation process in adult *Drosophila* midgut described in this study provides a genetically tractable system that should greatly facilitate the dissection of the genetic and molecular mechanisms underlying cell plasticity (Fig. 7h).

Our initial analysis with this in vivo transdifferentiation system reveals several insights into the molecular barriers to cell reprogramming (Fig. 7h). Firstly, chromatin accessibility may be a crucial factor that determines the feasibility and effectiveness of cell identity switches. Differentiated cells in a common stem cell lineage are more amenable to cell fate switches because they may share a permissive chromatin landscape. For instance, shared chromatin accessibility in mammalian ISCs, absorptive and secretory progenitors explains how secretory progenitors can adopt absorptive cell fate upon the loss of the master secretory cell fate regulator, atoh1[29]. This may also account for the ability of both types of intestinal progenitor cells and even differentiated cells to dedifferentiate back into ISCs following damage or nutrient fluctuation[4,47]. A recent study also shows that in newborn flies, EEs located in the anterior midgut have the ability to undergo nutrition-dependent cell dedifferentiation into ISCs[48]. Another example, alpha, beta, and delta cells in pancreatic islets, which share a similar chromatin accessibility landscape and are amenable for cell reprogramming towards beta cells with minimal genetic manipulations[49,50]. However, for cell identity switches across different germ layers, additional factors that remodel chromatin accessibility may become indispensable. When inducing pluripotent stem cells from fibroblasts, Oct4 and Sox2 in the Yamanaka factors are considered as pioneer factors because of their ability to make chromatin accessible in addition to their role in transcriptional regulation[51–53], and the ability to generate permissive chromatin landscapes may be key to promoting cell reprogramming and pluripotency induction by the Yamanaka factors.

Secondly, lineage conflict may represent a potential barrier to reprogramming. During the EC-to-EE transdifferentiation process, we found that the loss of Ttk not only induced expression of Pros, a major factor driving EE transdifferentiation but also upregulated Dpn, a factor involved in neuroblast self-renewal that inhibits cell differentiation. Depletion of Dpn significantly improved the efficiency of EC-to-EE transdifferentiation. It remains unclear why Dpn is activated following Ttk depletion, as the *dpn* locus is associated with closed chromatin status in both ISCs and differentiated intestinal cells[42]. Nevertheless, it is possible that during cell reprogramming by gene

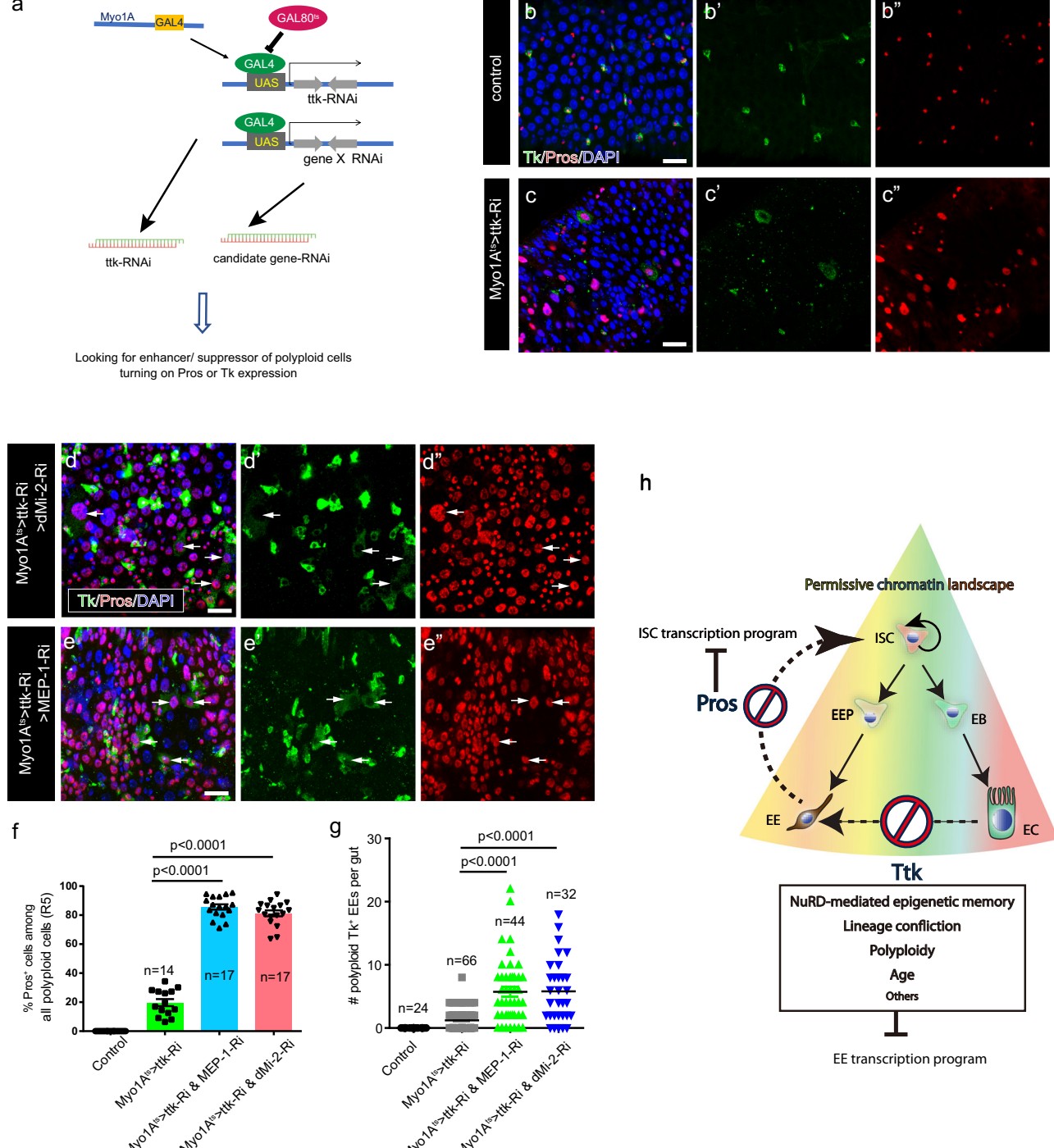

**Fig. 7 | Genetic screen identifying NuRD complex as a barrier to EC-to-EE transdifferentiation. a** Screening strategy for identifying potent suppressors of EC-to-EE trans-differentiation upon Ttk-depletion. Candidate TFs or epigenetic regulators were individually or co-depleted with Ttk-RNAi using Myo1A-GAL4. The percentage/number of polyploid cells expressing Pros or Tk was calculated and compared. **b, c** Compared to the control (**b**), knocking down ttk in ECs leads to the activation of Pros and Tk expression in a subset of these cells (**c**). **d, e** Compared to the control of Ttk knockdown alone (**c**), simultaneously knocking down Ttk with either MI2 (**d**) or MEP1 (**e**) significantly increases the proportion of polyploid cells expressing Pros and Tk. **f** Quantification of percentages of Pros[+] cells in polyploid cells. Error bars represent Mean ± SEM; ***$p < 0.001$ (two-tailed Student's $t$ test). "n" indicates the number of guts used for quantification; source data are provided as a Source Data file. **g** Quantification of the number of polyploid cells expressing Tk in each gut. Error bars represent Mean ± SEM ***$p < 0.001$ (two-tailed Student's $t$ test). "n" indicates the number of guts used for quantification; source data are provided as a Source Data file. **h** Schematic explaining cell-fate plasticity in the Drosophila ISC lineage. Despite having distinct transcriptome profiles, ISCs, EEs, and ECs exhibit a remarkably similar chromatin accessibility profile on loci associated with stem cell identity genes, EC identity genes, and EE identity genes. The permissive chromatin landscape allows for cell identity switches within the ISC lineage: Depletion of Ttk in ECs initiates EC-to-EE transdifferentiation, which is mediated by the master regulator Pros. Depletion of Pros in EEs initiates EE-to-ISC dedifferentiation due to the derepression of ISC identity genes. However, the reprogramming process is hindered by various barriers, including NuRD chromatin remodeling complex-mediated epigenetic memory, lineage confliction, polyploidy, and age, which limit fate-conversion efficiency. Scale bars, 20 μm.

manipulation with endogenous or exogenous factors, multiple lineage-promoting programs are activated simultaneously, leading to lineage confliction and hinders lineage commitment, thus constituting a barrier to transdifferentiation. Lineage confliction as a barrier to reprogramming has been indicated in several cases[40,41,54], and lineage confliction as a way to prevent lineage commitment has been considered as a strategy for the induction of pluripotency[55].

Lastly, our observations suggest that epigenetic memory may constitute another barrier to reprogramming. We noticed that young or newly-differentiated ECs had higher EC-to-EE transdifferentiation efficiency than older ECs. Young ECs were more amenable to Pros activation, and some even acquired the ability to produce peptide hormones, a hallmark of functional EEs. Additionally, our genetic screen showed that depletion of the NuRD chromatin remodeling complex significantly enhanced transdifferentiation efficiency, as nearly all ECs, including old ECs, had activated Pros. The NuRD complex, which possesses both chromatin remodeling and HDAC activities, has been implicated in Ttk-mediated transcriptional repression[44]. We suggest that the epigenetic modifications made by the NuRD complex at Ttk target loci could serve as an age-dependent epigenetic memory of the repressive chromatin status, thus hindering transcriptional de-repression of target genes upon Ttk depletion in elder ECs. Nevertheless, further studies with this system should lead to the identification of additional factors involved in the regulation of cell plasticity and cell reprogramming, which may eventually help to understand the molecular basis of the age factor that limits cell plasticity, a property that has important implications in diseases and regenerative medicine.

## Methods

### Fly strains and husbandry

The following fly strains were used in this study: Myo1A-GAL4[ts],UAS-GFP; Mex1-GAL4 (BDSC, #91368, 91369); GMR23G10-GAL4 (BDSC, #45845); prosV1-GAL4, UAS-GFP (gift from Bruce Edgar); Tk-GAL4[21]; Su(H)GBE (NRE)-GAL4 (gift from Steven Hou); UAS-ttk-RNAi#1(VDRC: GD4414); UAS-ttk-RNAi#2 (BDSC, #26315, TRiP.JF02088); UAS-pros-RNAi#1 (BDSC, #26745, TRiP.JF02308); UAS-pros-RNAi#2 (BDSC, #42538, TRiP.HMJ02107); UAS-E2f1-RNAi (BDSC, #27564, TRiP.JF02718); UAS-dpn-RNAi (BDSC, #26320, TRiP. JF02094); UAS-Mi-2-RNAi (BDSC, #33419, TRiP.HMS00301); UAS-MEP-1-RNAi (BDSC, #33676 TRiP.HMS00540); UAS-Pros (BDSC, #32244); UAS-ttk69 (BDSC, #7361); Rab3-EYFP (BDSC, #62541); esg-lacZ (BDSC, #10359); A142-GFP[56]; Tk-GFP[21]; Su(H)GBE-lacZ (NRE-lacZ, gift from Sarah Bray); Pros-mCherry-KI (generated in this study); and a recombined UAS-Flp,GAL80ts; Act<stop<lacZ, Tub-GAL80[ts] stock[16]. Fly stocks were cultivated on standard food with yeast paste added on the food surface and kept at 25 °C unless otherwise stated.

The GAL4/UAS/GAL80[ts] system was used to conduct conditional knocking down or overexpression of target genes in specific cell types[57,58]. Unless otherwise stated, all crosses were performed at 18 °C, and 5–7 day-old adult F1 progenies with correct genotypes were collected and transferred to 29 °C to induce gene expression.

### Generation of pros-mCherry-KI line

The Pros-mCherry-KI strain is a genetically modified line in which mCherry is inserted in-frame with the Pros gene. This was achieved by removing the stop codon of Pros and replacing it with a mCherry-w cassette, resulting in the production of a C-terminal mCherry-tagged Pros protein. The generation of the Pros-mCherry-KI line involved CRISPR/Cas9-mediated genome editing through homology-dependent repair (HDR). This process utilized a guide RNA and a double-stranded DNA plasmid donor. Specifically, the genomic DNA of the Pros locus was targeted and cut at the +4th nucleotide from the stop codon of the gene using the following guide RNA primers:

Sense oligo 5′-CTTCGAGAGCAGCTGGAATAAGTGG;
Antisense oligo 5′-AAACCCACTTATTCCAGCTGCTCTC

The donor vector utilized in this study contains a mCherry-white cassette, which is surrounded by upstream and downstream homology arms for facilitating homology-dependent recombination. The upstream homology arm spans 966 base pairs, ranging from −966 to −1 nucleotide positions relative to the stop codon of the pros gene. This homology arm was amplified using the following primers:

Forward oligo 5′- CCGACACCGACATATACACG
Reverse oligo 5′- TTCCAGCTGCTCTAAAAAATTG;

The downstream homology arm used in this study is 985 base pairs in length, spanning from +4 to +988 nucleotide positions relative to the stop codon of the pros gene. This homology arm was amplified using the following primers:

Forward oligo 5′- GTGGAGGAGTTGGCGCTG
Reverse oligo 5′- ATTCCCGATTTCCGTCCGTC

To introduce the desired genetic modifications, the sgRNA (single guide RNA) and the dsDNA donor vector were injected into the embryos. The white marker was employed as a selection marker to identify successful knock-in progeny. These selected progeny were then subjected to sequencing analysis to confirm the accurate incorporation of the knockin modification.

### Immunostaining

Immunostaining of the Drosophila midgut was conducted following previously described protocols[59]. Briefly, 8–10 guts from adult female flies were dissected in ice-cold 1X phosphate-buffered saline (PBS) and fixed with 4% paraformaldehyde for 30 min at room temperature. The samples were then dehydrated with methanol and rehydrated in a PBT solution (PBS containing 0.1% Triton X-100) for three cycles.

Next, the samples were incubated with primary antibodies diluted in a 5% normal goat serum (NGS)-PBT solution for 2 h at room temperature or overnight at 4 °C. After three washes with PBT, the samples were incubated with secondary antibodies for 2 h at room temperature. Nuclei were stained with DAPI for 5 min. The samples were mounted in 70% glycerol, and the slides were stored at −20 °C.

Imaging was performed using either the Nikon A1R or Leica SP8 confocal microscopes. All acquired images were adjusted and assembled using Adobe Photoshop and Illustrator. ImageJ was used for measuring fluorescence intensity, and for calculating nuclear size and cell number.

Primary antibodies used in this study were as follows: mouse anti-Pros (DSHB #MR1A; 1:300); mouse monoclonal anti-β-galactosidase (DSHB,# 40-1a; 1:30); mouse anti-Dl (DSHB Cat#C594.9B; RRID:AB_528194; 1:300); rabbit anti-AstC (lab generated antibody (RRID: AB_2753141) and gift from Dr. Dick Nassel; 1:300); rabbit anti-Tk (lab generated antibody (RRID: AB_2569591) and gift from Dr. Jan-Adrianus Veenstra; 1:300); rabbit polyclonal anti-β-galactosidase (Cappel, 0855976; 1: 3000); Rabbit anti-pH3 (CST Cat# 9701; RRID:AB_331535; 1:500); Rabbit anti-Pdm1 (lab generated antibody and gift from Dr. Xiaohang Yang; 1:200); Rabbit anti-Dpn (lab generated antibody and gift from Dr.Yuh-NungJan, RRID:AB_2567048;1:250); Rabbit anti-Sox100b (lab generated antibody in our lab; 1:500)[26]; Rabbit anti-Ttk69 (lab generated antibody in our lab; 1:200)[16]; Rabbit anti-Sox21a (lab generated antibody in our lab; 1:100)[36]. Secondary antibodies used in this study include Alexa Fluor 488-, 568- or Cy5-conjugated goat anti-rabbit, anti-mouse IgGs (Molecular Probes, A11034-A11036, A10524; 1:300). For nuclei staining, DAPI (Sigma-Aldrich, 1 μg/ml) was used.

### Fluorescence-activated cell sorting (FACS)

To obtain cells for sorting (for GFP+ and PI−), adult female flies aged 5–7 days with the following genotypes were cultivated at 29 °C for 7 days:

1. esg-GAL4, tubP-GAL80[ts], UAS-GFP/+
2. Myo1A-GAL4, UAS-GFP/+; tubP-GAL80[ts]/+
3. Myo1A-GAL4, Tk-GFP/ UAS-ttk-RNAi; tubP-GAL80[ts]/+
4. Myo1A-GAL4, UAS-GFP/ UAS-ttk-RNAi; tubP-GAL80[ts] / UAS-pros-RNAi

5. Myo1A-GAL4, UAS-GFP/+; tubP-GAL80ts / UAS-pros

6. UAS-GFP, tubP-GAL80ts/+; prosV1-GAL4/+

7. UAS-GFP, tubP-GAL80ts/+; prosV1-GAL4/ UAS-pros-RNAi

For flies with the genotype of Myo1A-GAL4, UAS-GFP/ UAS-ttk-RNAi; tubP-GAL80ts/ pros-mCherry-KI, GFPweak and mCherry+ cells were sorted.

FACS (Fluorescence-Activated Cell Sorting) was performed according to previously published protocols[60,61]. In brief, 100–150 guts for each sample were dissected in ice-cold DEPC-PBS (Diethylpyrocarbonate-treated phosphate-buffered saline) within 2 h, and the dissected guts were then digested in 1 mg/ml elastase solution (Sigma, cat. no. E0258) for 1 h at room temperature. The samples were centrifuged at 500 g for 10 min at 4 °C, and the cell pellet was resuspended in 600 µl of DEPC-PBS containing 1 µg/ml propidium iodide (PI, Invitrogen, #P3566), except for Myo1A-GAL4, UAS-GFP/UAS-ttk-RNAi; tubP-GAL80ts/ pros-mCherry-KI flies, where 1 µg/ml DAPI was used instead of PI. PI⁻ GFP⁺ (DAPI⁻; GFPweak; mCherry⁺) cells were sorted into ice-cold DEPC-PBS using a FACS Aria II sorter (BD Biosciences) equipped with FACSDiva software (Version 6.1.3) (Supplementary Fig. 10). The sorted cells were then used for subsequent RNA-sequencing and ATAC-sequencing experiments.

## RNA-sequencing and data analysis

FACS-sorted cells from the following genotypes were used for RNA-sequencing:

1. Myo1A-GAL4,UAS-GFP/+; tubP-GAL80ts/+

2. Myo1A-GAL4, UAS-GFP/ UAS-ttk-RNAi; tubP-GAL80ts/ UAS-pros-RNAi

3. Myo1A-GAL4, UAS-GFP/+; tubP-GAL80ts/ UAS-pros

4. Myo1A-GAL4, Tk-GFP/ UAS-ttk-RNAi; tubP-GAL80ts/+

For each sample, approximately 20,000 PI⁻ GFP⁺ cells were FACS sorted into ice-cold PBS and collected by centrifugation.

The RNA extraction and amplification steps were performed using the Arcturus PicoPure RNA isolation kit (Applied Biosystems, Cat#KIT0204) and the Arcturus RiboAmp HS PLUS RNA amplification kit (Applied Biosystems, Cat#KIT0525), respectively. The procedures were carried out following the manufacturer's instructions.

The amplified RNA samples were utilized for sequencing following previously established protocols[61]. Briefly, 1 µg of the amplified RNA samples were used for library preparation, and library construction was performed using the NEB Next Ultra II DNA library prep kits (New England Biolabs, cat. no. E7645L). Single-ended deep sequencing was conducted on an Illumina HiSeq-2500 sequencing system with a read length of 50 base pairs.

In addition to the datasets generated in this study, several previously published RNA-sequencing datasets from our lab were also included in the data analysis. These datasets were generated using the same methodology. The raw sequencing data underwent a filtering process, and only high-quality reads were retained. These filtered reads were then aligned to the D. melanogaster genome (BDGP6) using STAR (v2.7.10a), and counts were assigned to protein-coding genes using featureCounts (v2.0.1). DESeq2 was employed to identify significantly differentially expressed genes with the following parameters: adjusted p-value (Padj) < 0.01 and the absolute value of log2 fold change (FC) > 0.5. Gene Ontology (GO) analysis for the differentially expressed genes was performed using DAVID[62]. Heatmaps were generated using the R package "pheatmap".

In addition to the datasets generated in this study, bulk RNA-sequencing profiles of three cell types were utilized: enteroendocrine (EE) cells labeled by ProsV1-GAL4, progenitor cells labeled by esg-GAL4, and enterocyte (EC) cells labeled by Myo1A-GAL4. These profiles were previously described[15]. The top 500 genes with the highest expression scores were designated as cell type markers. These cell-type markers were then utilized as gene sets for the combined analysis of ATAC-seq and RNA-seq data.

## ATAC-sequencing and data analysis

For ATAC-sequencing, FACS-sorted cells from the following genotypes were used:

1. esg-GAL4, tubP-GAL80ts, UAS-GFP/+

2. Myo1A-GAL4,UAS-GFP/+; tubP-GAL80ts/+

3. UAS-GFP, tubP-GAL80ts/+; prosV1-GAL4/+

4. UAS-GFP, tubP-GAL80ts/+; prosV1-GAL4/ UAS-pros-RNAi

5. Myo1A-GAL4, UAS-GFP/UAS-ttk-RNAi; tubP-GAL80ts/ pros-mCherry-KI

Approximately 40,000 PI⁻ GFP⁺ (DAPI⁻; GFPweak; mCherry⁺) cells were FACS sorted for each sample. The sorted cells were collected by centrifugation into ice-cold PBS.

DNA tagmentation and library preparation were performed using the ATAC-Seq Kit (Active motif, Catalog No. 53150) following the manufacturer's instructions. Paired-end deep sequencing was conducted on an Illumina HiSeq-2500 sequencing system with a read length of 150 base pairs.

The raw sequencing data underwent preprocessing steps to eliminate low-quality reads and remove index sequences using Cutadapt software. The resulting clean data were then mapped to the D. melanogaster genome (BDGP6) using bowtie2 (version 2.3.5.1). PCR duplicates were removed using the picard MarkDuplicates function, while multimapped reads were eliminated using samtools. Regions listed in the blacklist were also removed using the bedtools intersect function. To generate bigWig (bw) files, the deepTools bamCoverage function was employed with BPM normalization. Finally, peaks were called using MACS3 to identify regions of enriched signal.

## Integrated transcriptome and chromatin accessibility analysis

Differential gene expression analysis was performed to identify genes that are differentially expressed between normal and Pros-depleted enteroendocrine cells (EEs), as well as between normal enterocytes (ECs) and transformed EEs. Additionally, cell type signature genes were determined using the methodology described earlier.

To evaluate the chromatin accessibility landscape of these genes, ATAC-seq peaks within a region spanning 3 kb upstream to 3 kb downstream of each gene locus were plotted. These peaks were ranked based on their openness, from strongest to weakest. The EnrichedHeatmap package was used to plot the ATACseq profile in combination with the transcriptome data.

To assess the similarity of transcriptome and ATAC-seq profiles among normal progenitor cells, EEs, ECs, and transformed cells, pairwise Pearson correlation analysis was performed.

The R package Diffbind (v3.10.0) was used to merge narrow peaks from all samples and to count & normalize the reads on peaks. Principal Component Analysis (PCA) plots were generated to visualize the similarity of chromatin accessibility among different cell types in the intestine, as well as among different organs.

## Statistics and reproducibility

All experiments were independently replicated at least 2–3 times, and consistent results were obtained. The manuscript includes representative figures that demonstrate the findings.

ImageJ software (version 1.48 v) was utilized for cell number counting. Quantitative data were presented as the mean ± standard error of the mean (SEM). Statistical analysis to determine significant differences was performed using GraphPad Prism 6 software (GraphPad Software Inc.). The p-values were calculated using the unpaired Student's t-test with Welch's correction.

## Reporting summary

Further information on research design is available in the Nature Portfolio Reporting Summary linked to this article.

## Data availability

The raw and processed datasets, including RNA-seq data and ATAC-seq results generated in this study, have been made available in the supplementary material or deposited in the GEO database under the accession code GSE235505. The genome datasets used in this study were BDGP6 for RNA-seq and ATAC-seq analysis. Additionally, three RNA-seq datasets previously reported by our lab are accessible in the GEO database under the following accession codes: GSE130943 (RNA-seq data of esg+ cell[42]), GSE130305 (RNA-seq data of ECs[26]), and GSE211632 (RNA-seq for control EE and Pros-depleted EE[15]). Furthermore, several public ATAC-seq datasets related to multiple tissues were utilized in this study: GSE157776 (intestinal stem cells[30]), GSE59078 (Drosophila eye-antennal discs from wandering third instar larvae[31]), SRX9186391-SRX9186393 (2-day-old Drosophila whole testis; https://www.ncbi.nlm.nih.gov/bioproject/PRJNA665509[63]), GSE81434 (OSC, ovarian somatic cells[32]), and GSE154645 (FACS sorted GFP+ neurons from larval brain, elav > Dcr-2, mCD8::GFP)[33]. Source data are provided with this paper.

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

## Acknowledgements

We thank Dr. Dick Nassel (Stockholm University) for providing anti-Tk antibody, Dr. Jan-Adrianus Veenstra (Université de Bordeaux) for anti-AstC antibody, Dr. Yuh-Nung Jan (UCSF) for anti-Dpn antibody and the Bloomington *Drosophila* Stock Center (BDSC), the VDRC stock center, the Tsinghua Fly Center, and Development Studies Hybridoma Bank (DSHB) for fly strains and antibodies. This work is supported by National Key Research and Development Program of China (2020YFA0803502 and 2017YFA0103602 to R.X.) from the Chinese Ministry of Science and Technology. X.G. is supported by National Natural Science Foundation of China (Grant No. 32100595).

## Author contributions

C.W. made the initial observation of EC-to-EE conversion and hence conceived as well as initiated the project. X.G., C.W. and R.X. designed the experiments. X.G., C.W. and R.W. performed the experiments. Y.Z. performed the bioinformatics analysis. R.X. and X.G. acquired fundings. X.G. and R.X. wrote the paper.

## Competing interests

The authors declare no competing interests.
