## [Peer Review File · Nature Communications]

Cell-fate conversion of intestinal cells in adult *Drosophila* midgut by depleting a single transcription factorREVIEWER COMMENTS

Reviewer #1 (Remarks to the Author):

This is an interesting manuscript that seeks to explore the regulation of lineage commitment and cellular plasticity in the *Drosophila* intestine. The manuscript is well written and the study is well-executed, using genetic strategies to assess how stable cellular identities are when terminally differentiated cells are perturbed by knock down of lineage specification transcription factors. The authors show that knocking down Ttk in ECs results in the acquisition of marker gene expression normally expressed in EEs, and that this is mediated by Pros upregulation, while depletion of Pros in EEs results in the acquisition of a proliferative phenotype that may be representative of ISC function. The authors further perform an ATACseq experiment that shows significant similarities in chromatin landscapes between ISCs, ECs, and EEs. Finally, the authors show that knockdown of TTK has more significant transdifferentiating effects in ECs when it happens early during EC maturation, and, using a genetic screen, identify components of the NURD complex as barriers to reprogramming.

While the study is interesting and addresses a critical question in the establishment and maintenance of lineage fidelity in adult stem cell populations, there are a number of shortcomings that would need to be resolved before the study can be recommended for publication:

1. The authors purely rely on RNAi approaches to perturb the system. This is problematic as it can result in incomplete loss of function, as well as in off-target effects. The authors need to include at least some experiments using validated loss of function alleles for Ttk and Pros to strengthen their conclusions.
2. It remains unclear whether Pros+ ECs are truly lineage converted or whether the knockdown of Ttk is simply de-repressing certain target genes that happen to be markers for EEs. The authors attempt to use EM as a way to bolster their claims, but it is unclear whether the EM sections shown really demonstrate conversion of cell states of ECs into a secretory phenotype (it also is not shown whether these particular cells are ECs, or Pros+, etc.). Further characterization of these cells, possibly using scRNAseq, would be necessary to clarify if these polyploid cells expressing Pros are really entering an EE-like state or just re-expressing Ttk-repressed genes.
3. The screen for chromatin modifiers influencing cell state transitions requires additional controls to make sure that changing chromatin states does not affect activity of the Gal4/UAS system used in the screen.

Reviewer #2 (Remarks to the Author):

In this manuscript, Guo et al., describe plasticity between EC and EE cell fates in the adult *Drosophila* midgut epithelium, which they find to be mediated by the transcription factor Tramtrack described by the Xi lab in previous works and a second transcription factor prospero. The putative cellular plasticity of terminally differentiated epithelial cells and the underlying transcriptional and chromatin profiles are of

high interest for our understanding of the underlying genetic and molecular mechanisms and for future therapeutical interventions.

Although the authors provide phenotypical and statistical evidence for their conclusions, I have major technical considerations and minor remarks that have to be addressed during revision.

Major

- Both kinds of plasticity experiments are based on solely one Gal4 driver each (EC-MyoIA> and EE-Pros>). As the authors indicate in Fig.1a and the Ohlstein lab showed as well, weak Pros activity is found in EEP, which would limit plasticity to precursor state (EEP).

Similar reports showed that late EB with GBE activity already possess Myo activity as well as the data the authors describe in Fig.6. High Sox21 is found in more Myo+ than in Mex+ cells supporting Myo-Gal4 activity in EB. The Uhlirova and Lemaitre labs described a role for Sox21 in EB differentiation before terminal EC fate. Additionally, several Pros+/Rab3+-cells in FigS1d (Myo>ttk-RNAi) also display horizontal protrusions resembling EB-like cells (Jasper and Dominguez labs), which should be at least discussed. Thus, at least key experiments, like Myo>ttk-RNAi and Pros>pros-RNAi, should be repeated with a second cell type specific Gal4 driver like Mex> and GME23G10> (Fig.5) and Rab3/Tk/AstC Gal4 lines to allow statements about terminally differentiated EC and EE cells and their plasticity.

- In line with this, lineage plasticity of progenitor cells leading to EE and EC fate changes have been observed previously by Jasper, Jones, Dominguez, Reiff and Jiang labs and are reviewed in Nagai et al., 2022. These publications should be discussed.

- Conclusions drawn about 'aged' EC and possibly associations with the drivers presented in Fig.6 need more detailed experimental approaches (Jasper, Jones, Partridge labs). A discrimination between 'young' in the sense of differentiation state (see my comment #1) or true age of EC needs to be thoroughly addressed experimentally to allow such conclusions.

- Additionally, polyploidy of EC has been involved to be a major obstacle to re-entry into cell cycle by Uhlirova, Lemaitre and Edgar labs and should be verified with additional drivers mentioned in comment #1.

Minor

- Fig. 1a weak not week.

- The authors switch between MyoIA and Myo1A (Fig.7), please follow Flybase nomenclature for all genes.

Reviewer #3 (Remarks to the Author):

This is an interesting study from Guo et al that investigates the potential of cell fate conversion in the adult Drosophila intestinal lineage, improving our understanding of the mechanisms that drive and maintain cell fate in this tissue. Their main conclusions are that loss of Ttk causes EC transdifferentiation into EEs, that loss of Pros induces dedifferentiation of EEs, and that this plasticity is due to these cells

sharing similar epigenetics landscape only for these transitions upon manipulation of single transcription factors.

While the experiments appear to meet expected standards in the field and support some of the conclusions, I believe it still falls a bit short of demonstrating the two main points: that transdifferentiation occurs following Ttk knock-down or that ISCs are generated in EE>prosRNAi. In both cases potential caveats need to be addressed. First, can the authors rule out that their “young ECs drivers” are not active in enteroblasts and that what they call young ECs in Fig6 are not what others have reported as late EBs. While this may seem only semantic, it is critical as it would change the interpretation from true transdifferentiation to a rather less unexpected shift in differentiation path. It would also challenge the conclusion that EC age represents a barrier for cell conversion, but rather that merely EC differentiation does. Second, additional markers and assays are required to definitely demonstrate that the dividing cells generated by loss of Pros in EEs are true ISCs. This is critical to ensure that the cells analyzed throughout this work are indeed converted, and not aberrant cell phenotypes induced by genetic perturbations.

Specific points:

- What is the overlap between the activity of the EC drivers and the EB marker NRE-LacZ? Does TtkRNAi expression with the GBE-Su(H)::Gal4 driver result in the formation of polyploid pros+ cells?
- The experiment presented in Fig4p,q should include Pdm1 immunostaining as a marker of EC differentiation, as opposed to apparent polyploidy. This is essential to support the conclusion that dedifferentiated EEs into ISCs can then differentiate into ECs.
- How do the authors know in Fig2f that they are pointing at a “dedifferentiated pros+ cells” and not a normal EE?
- Additional PCA analysis should be performed without the other tissues presented in Fig5a. With these included, it is not possible to visualize how similar the “converted” cell types are compared to normal ISC, EE and ECs.
- Can the authors provide some statistical analysis to back up their claims that ATACseq profiles are largely unchanged? This is particularly necessary for the data presented in Fig5g.

Minor point:

- The mention of 4585-gal4 on page 17 appears to refer to GMR23G10
- How were cells categorized as polyploid for the quantification of Pros+ transdifferentiated cells?

We express our sincere gratitude to all the reviewers for dedicating their valuable time to carefully read our manuscript and provide insightful feedback. We greatly appreciate their constructive comments, which have helped us improve the quality of our work. In this revised manuscript, we have taken into account all the raised comments and made significant changes accordingly. Some of the major modifications implemented are as follows:

1. We have provided additional experimental evidence to support the occurrence of EC-to-EE transdifferentiation rather than EB-to-EE transdifferentiation. In our previous study, the depletion of Ttk in ECs was primarily achieved using the Myo1A-GAL4 driver. However, since Myo1A-GAL4 also exhibits leaky expression in EBs, there was a concern that the observed phenotype might be attributed to a shift in differentiation from EBs to EEs. To address this concern, we employed two additional EC-specific drivers (mex1-GAL4 and GMR23G10-GAL4) that do not exhibit leaky expression in EBs. Depletion of Ttk in ECs using these drivers still resulted in an EC-to-EE transdifferentiation phenotype. Furthermore, we used Su(H)-GBE-GAL4 (NRE-GAL4) to drive Ttk depletion specifically in EBs. This led to the differentiation of all EBs into diploid EEs, with no generation of polyploid Pros⁺ cells. These additional experiments provide compelling evidence supporting our conclusion that EC-to-EE transdifferentiation occurs upon Ttk depletion in ECs.
2. We have also provided additional experimental evidence to support the occurrence of EE-to-ISC dedifferentiation rather than EEP-to-ISC dedifferentiation. Previously, we primarily used the pros-GAL4 driver to deplete Pros in EEs. However, as ProsV1-GAL4 is also expressed in EEPs, there was a concern that the observed dedifferentiation phenotype might be a result of dedifferentiation from early progenitor cells (EEPs). To address this concern, we employed another EE-specific driver, Tk-GAL4, for the dedifferentiation experiment. Interestingly, we observed the emergence of DI⁺ cells from lineage⁺ cells following Pros depletion, and with a prolonged tracing period of 7 days, we observed the emergence of polyploid cells expressing Pdm1 from lineage⁺ cells as well. These additional data strongly support our conclusion that depletion of Pros leads to EE-to-ISC transdifferentiation.

In addition, we have presented further evidence to support the direct transdifferentiation from ECs to EEs, without an intermediate progenitor cell stage, and included new results to indicate that polyploidy could

serve as a potential barrier to the EC-to-EE transdifferentiation. We addressed the specificity and effectiveness of *ttk*-RNAi or *pros*-RNAi effects as well. Furthermore, we have made improvements to the language throughout the manuscript, reorganized the panels for better clarity, unified the GAL4 line names for consistency, and added suggested discussions to enhance the overall presentation of our findings.

Please find our point-by-point responses below.

Reviewer #1 (Remarks to the Author):

This is an interesting manuscript that seeks to explore the regulation of lineage commitment and cellular plasticity in the *Drosophila* intestine. The manuscript is well written and the study is well-executed, using genetic strategies to assess how stable cellular identities are when terminally differentiated cells are perturbed by knock down of lineage specification transcription factors. The authors show that knocking down *Ttk* in ECs results in the acquisition of marker gene expression normally expressed in EEs, and that this is mediated by *Pros* upregulation, while depletion of *Pros* in EEs results in the acquisition of a proliferative phenotype that may be representative of ISC function. The authors further perform an ATACseq experiment that shows significant similarities in chromatin landscapes between ISCs, ECs, and EEs. Finally, the authors show that knockdown of *TTK* has more significant transdifferentiating effects in ECs when it happens early during EC maturation, and, using a genetic screen, identify components of the NURD complex as barriers to reprogramming.

We thank the reviewer for these positive remarks.

While the study is interesting and addresses a critical question in the establishment and maintenance of lineage fidelity in adult stem cell populations, there are a number of shortcomings that would need to be resolved before the study can be recommended for publication:

1. The authors purely rely on RNAi approaches to perturb the system. This is problematic as it can result in incomplete loss of function, as well as in off-target effects. The authors need to include at least some experiments using validated loss of function alleles for *Ttk* and *Pros* to strengthen their conclusions.

Thanks for the suggestions. While we acknowledge that RNAi approaches can lead to incomplete gene function loss and off-target effects, it was not feasible for us to utilize loss-of-function alleles and perform clonal analysis to analyze the biological functions of Pros and Ttk in post-mitotic EE and ECs. This is because clonal induction relies on cell mitosis, which is not applicable in this case.

However, there is substantial evidence supporting that the observed phenotypes are a result of Pros or Ttk depletion in ECs:

1. Immunostaining revealed a nearly complete loss of Ttk69 staining in GFP⁺ ECs upon *ttk*-RNAi (Rebuttal Figure 1a-b), and a complete loss of Pros staining in GFP⁺ EEs upon *pros*-RNAi (Rebuttal Figure 1c-d). This indicates the effectiveness of these RNAi lines.

Rebuttal Figure 1. Ttk-RNAi and pros-RNAi lines are effective in depleting Ttk (a-b) and Pros (c-d) expression.

2. Both *ttk*-RNAi and *pros*-RNAi were performed using two independent RNAi lines with distinct hairpin sequences (VDRC: GD4414 and BDSC: TRiP.JF02088 for *ttk*-RNAi, and BDSC: TRiP.JF02308 and TRiP.HMJ02107 for *pros*-RNAi). Remarkably, both RNAi lines resulted in the same phenotype, suggesting that the observed phenotypes are not due to off-target effects (Figure 1c,d & Figure 2b,c,e,f for *ttk*-RNAi, and Figure 4 & Supplementary Figure 6 for *pros*-RNAi).
3. To further validate the role of Ttk and Pros in ECs and EEs, respectively, we co-expressed UAS-*ttk69* and in ECs driven by Myo1A-GAL4, UAS-*ttk*-RNAi as well as UAS-*pros* and in EEs driven

by prosV1-GAL4, UAS-pros-RNAi. As shown in the Figures below, overexpressing Ttk69 or Pros successfully rescued Ttk and Pros depletion caused by ttk-RNAi or pros-RNAi (Rebuttal Figure 2), along with the corresponding EC-to-EE transition or EE dedifferentiation phenotypes (Rebuttal Figure 3). These evidences rule out the possibility of off-target effects.

Rebuttal Figure 2. Overexpressing Ttk69 prevents Ttk depletion caused by ttk-RNAi in ECs (a); and overexpressing Pros prevents Pros depletion caused by pros-RNAi in EEs (b).

Rebuttal Figure 3. Overexpressing Ttk69 prevents Ttk depletion- induced Pros activation (a) and down regulation of Myo1A>GFP and Pdm1 expression (b); overexpressing Pros prevents Pros depletion-induced EE-to-ISC dedifferentiation(c).

2. It remains unclear whether Pros⁺ ECs are truly lineage converted or whether the knockdown of Ttk is simply de-repressing certain target genes that happen to be markers for EEs. The authors attempt to use EM as a way to bolster their claims, but it is unclear whether the EM sections shown really demonstrate conversion of cell states of ECs into a secretory phenotype (it also is not shown whether these particular cells are ECs, or Pros⁺, etc.). Further characterization of these cells, possibly using scRNAseq, would be necessary to clarify if these polyploid cells expressing Pros are really entering an EE-like state or just re-expressing Ttk-repressed genes.

Thanks for the suggestions. We are confident that the Pros⁺ ECs observed after Ttk-depletion underwent true lineage conversion rather

than simply de-repressing target genes coincidentally associated with EE markers. We base this assertion on the following arguments:

1. The Pros⁺ ECs exhibited a global shift in transcriptome towards EE identity, encompassing not only EE-specific transcription factors, peptide hormones, GPCRs, and secretive process-related genes but also a comprehensive loss of EC identity gene expression, including EC functional-related genes such as digestive enzymes (Figure 2i-j, Supplementary Figure 1e-f).
2. By examining the presence of the Ttk binding motif AGGGC/TGG (PubMed:8247159) within a -2kb to +2kb region of the upregulated genes following Ttk-depletion, we found that a significant proportion of these genes (approximately 75%) did not contain the Ttk binding motifs. This indicates that the activation of these genes cannot be solely attributed to the transient de-repression of Ttk target genes.
3. When we reverted EC>ttk-RNAi flies back to 18°C culture after inducing Tk⁺ polyploid cells for 7 days, we observed the persistence of these cells. This supports the induction of a stable transformed cell state rather than a transient activation of Ttk-suppressed target gene expression (Supplementary Figure 5c).
4. The transformed cells displayed diverse expression patterns of peptide hormones across different cells. Immunostaining confirmed the mutually-exclusive expression of Tk and AstC in polyploid Pros⁺ cells, resembling the pattern observed in normal EEs (Supplementary Figure 5a-b).

In Fig. 2f, we present an electron microscopy (EM) image where we selected one representative cell. However, we examined numerous EM images of the Myo1A>ttk-Ri gut, and consistently observed polyploid cells exhibiting an abundance of secretory vesicles in the cytoplasm. This phenomenon was not observed in the normal gut. Given the frequent occurrence of these cells, we strongly believe that they correspond to the Pros⁺ cells in the Myo1A>ttk-Ri gut. Notably, despite acquiring a secretory vesicle phenotype, these transformed cells still retain several morphological features characteristic of enterocytes, including polyploidy, a columnar cell shape, and the presence of microvilli on their apical membrane. These observations suggest that the transformed cells, while gaining secretory function similar to EEs, maintain many of the morphological characteristics of enterocytes.

We acknowledge the potential benefits of performing single-cell analysis to characterize the intermediate cell states in the process of EC-to-EE transdifferentiation. However, the size of these cells poses a limitation for single-cell analysis on the 10X Genomics platform, which is currently the only accessible single-cell platform for our research. To gain further insights into the process of EC-to-EE transdifferentiation, we conducted an examination of *esg-lacZ* expression, a marker for progenitor cells, at various time points following *Ttk*-depletion induction. We found that polyploid cells consistently remained negative for this marker (Supplementary Figure 2). These findings, in conjunction with the gradual decrease in *Pdm1* expression accompanied by an increase in *Pros* levels, as well as the coexistence of *Pros*⁺ and *Pdm1*⁺ cells during the transdifferentiation process, strongly support the direct nature of EC-to-EE transdifferentiation without involving a dedifferentiation step (Figure 1h-i). Furthermore, when *EC>ttk-RNAi* flies were shifted back to a culture temperature of 18°C for 7 days, the transdifferentiated cells persisted (Supplementary Figure 5c), indicating the stability of the acquired cell fate through transdifferentiation. This observation is not consistent with the idea that these cells are merely transiently activating *Ttk*-suppressed genes and provides evidence that they have genuinely undergone transdifferentiation.

3. The screen for chromatin modifiers influencing cell state transitions requires additional controls to make sure that changing chromatin states does not affect activity of the *Gal4/UAS* system used in the screen.

Thanks for the suggestion. We acknowledge that manipulating the expression of epigenetic regulators may impact *GAL4/UAS* activity. To assess whether the targeted epigenetic regulators could affect *Myo1A-GAL4* activity, we utilized *Myo1A-GAL4; UAS-GFP* to drive the RNAi lines of the two identified positive hits, *MEP1* and *dMi-2*. We compared the GFP levels with those of the control *Myo1A-GAL4; UAS-GFP*. As illustrated below, the results demonstrate that neither *dMi-2* nor *MEP-1* altered the activity of the *GAL4/UAS* system.

Rebuttal Figure 4: The GFP levels remained unchanged following the depletion of dMi-2 or MEP1.

Regarding all the other epigenetic regulators that were screened in this study, we recognize the concern that the depletion of specific epigenetic regulators could potentially impair the activity of Myo1A-Gal4, leading to false negative results.

Reviewer #2 (Remarks to the Author):

In this manuscript, Guo et al., describe plasticity between EC and EE cell fates in the adult *Drosophila* midgut epithelium, which they find to be mediated by the transcription factor Tramtrack described by the Xi lab in previous works and a second transcription factor prospero. The putative cellular plasticity of terminally differentiated epithelial cells and the underlying transcriptional and chromatin profiles are of high interest for our understanding of the underlying genetic and molecular mechanisms and for future therapeutical interventions.

We thank the reviewer for these positive remarks.

Although the authors provide phenotypical and statistical evidence for their conclusions, I have major technical considerations and minor remarks that

have to be addressed during revision.

Major

- Both kinds of plasticity experiments are based on solely one Gal4 driver each (EC-Myo1A> and EE-Pros>). As the authors indicate in Fig.1a and the Ohlstein lab showed as well, weak Pros activity is found in EEP, which would limit plasticity to precursor state (EEP).

Similar reports showed that late EB with GBE activity already possess Myo activity as well as the data the authors describe in Fig.6. High Sox21 is found in more Myo+ than in Mex+ cells supporting Myo-Gal4 activity in EB. The Uhlirova and Lemaitre labs described a role for Sox21 in EB differentiation before terminal EC fate. Additionally, several Pros+/Rab3+-cells in FigS1d (Myo>ttk-RNAi) also display horizontal protrusions resembling EB-like cells (Jasper and Dominguez labs), which should be at least discussed.

Thus, at least key experiments, like Myo>ttk-RNAi and Pros>pros-RNAi, should be repeated with a second cell type specific Gal4 driver like Mex> and GME23G10> (Fig.5) and Rab3/Tk/AstC Gal4 lines to allow statements about terminally differentiated EC and EE cells and their plasticity.

Thanks for the suggestions. As we summarized at the beginning of this rebuttal letter, we have employed additional drivers and provided additional experimental support for EC-to-EE transdifferentiation instead of EB-to-EE transdifferentiation, and for EE-to-ISC dedifferentiation instead of EEP-to-ISC dedifferentiation.

For the process of EC-to-EE transdifferentiation, we employed two additional EC-specific GAL4 drivers, namely mex1-GAL4 and GMR23G10-GAL4. Similarly, for the EE-to-ISC dedifferentiation, we employed Tk-GAL4 as an additional EE-specific driver. In all of these cases, we cell conversion phenotypes were similarly observed.

- In line with this, lineage plasticity of progenitor cells leading to EE and EC fate changes have been observed previously by Jasper, Jones, Dominguez, Reiff and Jiang labs and are reviewed in Nagai et al., 2022. These publications should be discussed.

We have now added the discussion to these observations in the introduction of the manuscript.

- Conclusions drawn about 'aged' EC and possibly associations with the drivers presented in Fig.6 need more detailed experimental approaches (Jasper, Jones, Partridge labs). A discrimination between 'young' in the sense of differentiation state (see my comment #1) or true age of EC needs to be thoroughly addressed experimentally to allow such conclusions.

Thanks for the suggestions. We find that it is challenging to define young and aged ECs. Initially, our intention in using terms like "young" or "aged" was to differentiate between newly formed and preexisting ECs, which are challenging to define as well. However, to alleviate any potential confusion, we have introduced alternative terminology in relevant instances to ensure clarity and avoid ambiguity.

- Additionally, polyploidy of EC has been involved to be a major obstacle to re-entry into cell cycle by Uhlirova, Lemaitre and Edgar labs and should be verified with additional drivers mentioned in comment #1.

Thanks for the suggestions. It is known that ECs in the fly midgut undergo endoreplication as they differentiate (source: PMID: 28485389). In our study, we found that depleting the Ttk gene did not seem to cause cells to re-enter the cell cycle. However, it did raise an interesting question about whether polyploidy could hinder the process of cell transdifferentiation.

Previous studies have identified the transcriptional factor E2F1 as a crucial regulator of endoreplication (sources: PMID: 28485389 and 22037307). When we knocked down the E2F1 gene, we observed a significant reduction in the nuclear sizes of myo1A>GFP⁺ cells (see supplementary Figure 7a-b), indicating an interference with EC polyploidy. Furthermore, depleting both ttk and E2F1 genes led to the generation of polyploid Pros⁺ cells at a rate of 32.6±3.0%, which was significantly higher than the rate observed with ttk depletion alone (18.4±2.8%) (see supplementary Figure 7c-d). These findings suggest that transdifferentiation can occur in polyploid ECs, but polyploidy may present an obstacle to transdifferentiation. We have included these new results in the manuscript.

Minor

- Fig. 1a weak not week.

Corrected.

- The authors switch between Myo1A and Myo1A (Fig.7), please follow Flybase nomenclature for all genes.

Corrected and checked thoroughly for similar cases.

Reviewer #3 (Remarks to the Author):

This is an interesting study from Guo et al that investigates the potential of cell fate conversion in the adult *Drosophila* intestinal lineage, improving our understanding of the mechanisms that drive and maintain cell fate in this tissue. Their main conclusions are that loss of Ttk causes EC transdifferentiation into EEs, that loss of Pros induces dedifferentiation of EEs, and that this plasticity is due to these cells sharing similar epigenetics landscape only for these transitions upon manipulation of single transcription factors.

While the experiments appear to meet expected standards in the field and support some of the conclusions, I believe it still falls a bit short of demonstrating the two main points: that transdifferentiation occurs following Ttk knock-down or that ISCs are generated in EE>prosRNAi. In both cases potential caveats need to be addressed. First, can the authors rule out that their “young ECs drivers” are not active in enteroblasts and that what they call young ECs in Fig6 are not what others have reported as late EBs. While this may seem only semantic, it is critical as it would change the interpretation from true transdifferentiation to a rather less unexpected shift in differentiation path. It would also challenge the conclusion that EC age represents a barrier for cell conversion, but rather that merely EC differentiation does. Second, additional markers and assays are required to definitely demonstrate that the dividing cells generated by loss of Pros in EEs are true ISCs. This is critical to ensure that the cells analyzed throughout this work are indeed converted, and not aberrant cell phenotypes induced by genetic perturbations.

We thank the reviewer for their positive remarks on this work as well as their constructive comments.

As we summarized at the beginning of this rebuttal letter, we have provided additional experimental support for EC-to-EE transdifferentiation instead of EB-to-EE transdifferentiation. In the previous study, we primarily used the Myo1A-GAL4 driver to deplete Ttk in ECs, but as Myo1A-GAL4 also has leaky expression in EBs (it is indeed the case, see supplementary Figure 3a-c), it raises a concern that the observed phenotype may be simply a result of shift in differentiation path of progenitor cells to EE identity. We employed two additional EC-specific drivers (mex1-GAL4 and GMR23G10-GAL4), which have no leaky expression in EBs, to deplete Ttk in ECs, and a similar EC-to-EE transdifferentiation phenotype was observed (Figure 6 and Supplementary Figure 4). We also employed Su(H)-GBE-GAL4 (NRE-GAL4) to drive Ttk depletion in EBs. This causes all EBs to differentiate into diploid EEs, and no polyploid Pros⁺ cells could be generated (Supplementary Figure 3d-e). Collectively, these results strongly support our conclusion that EC-to-EE transdifferentiation

occurs following Ttk depletion in ECs.

We fully acknowledge the challenges in defining terms such as young/aged and new/old when referring to ECs, which may appear to be purely semantic distinctions. To prevent any confusion, we have introduced alternative terminology whenever necessary to reflect this issue.

To further confirm that the dividing cells induced by Pros-depletion in EEs are indeed true ISCs, we conducted co-staining of PH3 with DI and verified that these PH3⁺ cells also expressed DI. Additionally, we employed another mature EE driver, Tk-GAL4, and observed a similar dedifferentiation phenotype. Notably, the resulting ISCs were capable of dividing and generating Pdm1⁺ ECs, demonstrating their ISC functionality (Figure 4s-u). Furthermore, as shown in Figure 4a-c, we observed a general upregulation of progenitor identity genes following Pros-depletion in EEs. Collectively, these findings strongly support the notion that Pros depletion in EEs triggers cell dedifferentiation and the acquisition of functional ISCs.

Specific points:

- What is the overlap between the activity of the EC drivers and the EB marker NRE-LacZ? Does TtkRNAi expression with the GBE-Su(H)::Gal4 driver result in the formation of polyploid pros⁺ cells?

Thanks for the suggestions. In total, we have employed three specific drivers for ECs: Myo1A-Gal4, mex1-GAL4, and GMR23G10-GAL4. Through co-staining with the EB marker NRE-lacZ, we observed that Myo1A-GAL4 exhibited some leakage of expression in EBs, whereas mex1-GAL4 and GMR23G10-GAL4 were specific to ECs (Supplementary Figure 3a-c). Importantly, we observed that the EC-to-EE transdifferentiation could similarly be induced using either mex1-GAL4 or GMR23G10-GAL4 as drivers (Figure 6 and Supplementary Figure 4).

To induce Ttk depletion in EBs, we employed Su(H)-GBE-GAL4 (NRE-GAL4) as the driver. Following Ttk-depletion, we observed an accumulation of Pros⁺ diploid EEs, but no polyploid Pros⁺ cells were detected. Furthermore, the results from the cell lineage tracing experiment demonstrated that EBs with Ttk depletion exclusively gave rise to Pros⁺ diploid cells (Supplementary Figure 3d-e). These observations strongly support our conclusion that the reprogrammed polyploid cells originate from ECs rather than EBs.

- The experiment presented in Fig4p,q should include Pdm1 immunostaining as a marker of EC differentiation, as opposed to apparent polyploidy. This is essential to support the conclusion that dedifferentiated EEs into ISCs can then differentiate into ECs.

Done as suggested. Our findings demonstrate that the depletion of Pros in EE lineage tracing, driven by either ProsV1-GAL4 or Tk-GAL4, resulted in the emergence of DI+ cells within their respective lineages. Furthermore, prolonged tracing led to the generation of multicellular clones that included Pdm1+ ECs (Figure 4r, u)

- How do the authors know in Fig2f that they are pointing at a “dedifferentiated pros+ cells” and not a normal EE?

We classify the cell indicated in Figure 2f as a transdifferentiated EC rather than a normal EE for the following reasons: Firstly, the columnar shape of the cell is consistent with the shape of normal ECs, while differing from the commonly observed spindle- or triangle-shaped morphology of normal EEs. Secondly, the cell retains a large luminal surface area and microvillous structures, despite undergoing cell fate reprogramming. In contrast, normal EEs typically have a small apical surface without microvillous structures (Ruei-Jiun Hung et al., PNAS, 2020). Therefore, we define the cell highlighted in Figure 2f as a transdifferentiated cell originating from an EC.

- Additional PCA analysis should be performed without the other tissues presented in Fig5a. With these included, it is not possible to visualize how similar the “converted” cell types are compared to normal ISC, EE and ECs.

Thanks for the suggestion. We removed public ATAC-seq profiles of other tissues, while still reserved the 3 public ATAC-seq profiles of intestinal stem cells (GSE157776; PMID: 33724181) to generate the PCA plot with our ATAC-seq profiles. As is shown below in Rebuttal Figure 5, the distribution landscape for these cell clusters is largely similar to the one shown in Figure 5a. The EC-to-EE transdifferentiated cells remained relatively close to the EC cluster and the EE-to-ISC dedifferentiated cells remained relative close to the EE cluster. However, we acknowledge that there may be experimental variations, particularly when comparing data generated from different laboratories, which could potentially affect the interpretation of the results.

Rebuttal Figure 5. PCA plot for ATAC-seq of different intestinal cell types.

- Can the authors provide some statistical analysis to back up their claims that ATACseq profiles are largely unchanged? This is particularly necessary for the data presented in Fig5g.

We took the suggestion and conducted a Pearson correlation analysis of ATAC-seq profiles comparing normal cells to transformed cells, with a focus on genomic regions associated with genes that exhibited significant transcriptional changes (as shown in the left panel of Figure 5f and 5g). These genes displayed a significant negative correlation at the transcriptional level. However, when examining the effect of pros-depletion on these altered genes, we observed a positive correlation score of 0.89 between the ATAC-seq profiles of normal and pros-depleted EEs. Additionally, there was a positive correlation score of 0.65 between the ATAC-seq profiles of the altered genes in normal cells and *ttk*-IR-induced fate converted ECs. These findings support the notion that the ATAC-seq profiles of these significantly altered genes remained largely unchanged during the process of reprogramming.

Minor point:

- The mention of 4585-gal4 on page 17 appears to refer to GMR23G10

Yes, corrected.

- How were cells categorized as polyploid for the quantification of Pros+ transdifferentiated cells?

As shown in Supplementary Figure 1a, we assessed the ploidy of

individual cells by measuring their nuclear sizes using Image J software. In the case of normal EEs, their nuclear sizes typically fall below 20 μm^2 . Conversely, for the Pros⁺ transdifferentiated cells, we designated nuclear sizes exceeding 20 μm^2 as indicative of polyploid cells. We have included this information in the manuscript.

REVIEWERS' COMMENTS

Reviewer #1 (Remarks to the Author):

The authors have included additional data, controls and further explanations in their revised version that now satisfy my concerns. This is an interesting study that can be recommended for publication.

Reviewer #2 (Remarks to the Author):

In this revised manuscript, Guo et al. address all major points raised by my initial review and also by two more reviewers. During the revision process, the authors added important data with further established cell-type specific drivers and added to their discussion of existing literature. Both additions substantially improved evidence and significance of their main findings and the understanding of the obtained results. In addition, the authors also discuss polyploidy now with new data involving E2F1 mediated endoreplication as obstacle for transdifferentiation. Finally, i want to congratulate the authors for this important contribution to our understanding of cellular plasticity in the intestinal epithelium.

Reviewer #3 (Remarks to the Author):

In this revised version of their manuscript, Guo et al. provide additional evidence supporting the main conclusions of their study. They present data to directly and successfully address my specific comments and most concerns expressed by other reviewers. Overall, this work strengthens our understanding of cellular plasticity in the Drosophila intestinal lineage.